# Regional Greenland Accumulation Variability from Operation IceBridge Airborne Accumulation Radar

Lewis, Gabriel[1]; Osterberg, Erich[1]; Hawley, Robert[1]; Whitmore, Brian.[1].; Marshall, Hans Peter[2]; Box, Jason[3]

[1]Department of Earth Sciences, Dartmouth College, Hanover, NH, USA
[2]Geosciences Department, Boise State University, Boise, ID, USA
[3]Geological Survey of Denmark and Greenland (GEUS), Copenhagen, Denmark

*Correspondence to*: Gabriel Lewis (Gabriel.M.Lewis.GR@dartmouth.edu)

**Abstract.** The mass balance of the Greenland Ice Sheet (GrIS) in a warming climate is of critical interest to scientists and the general public in the context of future sea-level rise. An improved understanding of temporal and spatial variability of snow accumulation will reduce uncertainties in GrIS mass balance models and improve projections of Greenland's contribution to sea-level rise, currently estimated at 0.089 ± 0.03 m by 2100. Here we analyze 25 NASA Operation IceBridge Accumulation Radar flights totaling >17,700 km from 2013-2014 to determine snow accumulation in the GrIS dry snow and percolation zones over the past 100-300 years. IceBridge accumulation rates are calculated and used to validate accumulation rates from three regional climate models. Averaged over all 25 flights, the RMS difference between the models and IceBridge accumulation is between 0.023 ± 0.019 and 0.043 ± 0.029 m w.e. a$^{-1}$, although each model shows significantly larger differences from IceBridge accumulation on a regional basis. In the southeast region, for example, the Modèle Atmosphérique Régional (MARv3.5.2) overestimates by an average of 20.89 ± 6.75% across the drainage basin. Our results indicate that these regional differences between model and IceBridge accumulation are large enough to significantly alter GrIS surface mass balance estimates. Empirical orthogonal function analysis suggests that the first two principal components account for 33% and 19% of the variance and correlate with the Atlantic Multidecadal Oscillation (AMO) and wintertime North Atlantic Oscillation (NAO), respectively. Regions that disagree strongest with climate models are those in which we have the fewest IceBridge data points, requiring additional *in situ* measurements to verify model uncertainties.

## 1. Introduction

Assessing the stability of the Greenland Ice Sheet (GrIS) in a warming world is crucial for predicating future global sea-level rise and its societal and economic impacts (Dumont et al., 2014; IPCC, 2014). The mass balance of the GrIS has been decreasing over the 1988-2016 period, with a conservative estimate of ice sheet mass loss of 272 ± 24 Gt a$^{-1}$ (van den Broeke et al., 2016; Enderlin et al., 2014; Hanna et al.,

2013a; Khan et al., 2015; Sasgen et al., 2012; Shepherd et al., 2012), or an equivalent global sea-level rise of ~0.7 ± 0.2 mm a$^{-1}$ (Ettema et al., 2009; Helm et al., 2014). The dominant mass loss process for the GrIS has changed from ice discharge (i.e. calving) to surface mass balance (SMB) since the mid-1990s (van den Broeke et al., 2009, 2016). SMB is one of the largest sources of error in estimates of the ice sheet's total

mass balance (van den Broeke et al., 2009) due to complex relationships between accumulation variability and surface melt runoff (Dumont et al., 2014; Hanna et al., 2005; McConnell et al., 2000). GrIS snow accumulation varies spatially in response to surface topography (e.g. Hawley et al., 2014), wind redistribution (Déry and Yau, 2002), and preferred modes of climate variability like the North Atlantic Oscillation (NAO; e.g. Wong et al., 2015), Atlantic Multidecadal Oscillation (AMO; e.g. Mernild et al.,

2014), and Greenland Blocking Index (GBI; e.g. Hanna et al., 2016). Accumulation also varies through time largely in response to temporal changes in these climate modes (Mernild et al., 2014). Ice cores accurately record temporal accumulation changes at point locations (Banta and McConnell, 2007; McConnell et al., 2000; Mosley-Thompson et al., 2001), and have been used with snow pits and coastal precipitation data to determine large-scale accumulation patterns over the entire ice sheet (Bales et al.,

2009). However, ice cores and snow pits are too sparse to capture the full spatial variability of GrIS accumulation, especially in the high-accumulation southeast region where little *in situ* data exists. Further, many Greenland ice cores were collected during the 1990s or earlier, prior to the recent acceleration of GrIS mass loss (Box et al., 2013). An updated, more spatially distributed, and representative GrIS accumulation dataset is needed to evaluate recent precipitation trends and to validate GrIS SMB estimates

from regional climate models (RCMs) over recent decades of increased mass loss.

Here we develop a record of GrIS snow accumulation over a large portion of the GrIS interior from 1712 to 2014 AD using the airborne NASA Operation IceBridge Accumulation Radar (Leuschen et al., 2011). Airborne and ground-based radars have been used to map spatial patterns of accumulation in Greenland

over decadal (Hawley et al., 2014; Miège et al., 2013) and annual resolutions (Koenig et al., 2016; Medley et al., 2013). Operation IceBridge collected Accumulation Radar data from 2009-2014, and it has been used in several studies (Karlsson et al., 2016; Forster et al., 2014; Leuschen et al., 2011; Medley et al., 2013) to calculate local accumulation. We examine Accumulation Radar data from every IceBridge flight across the Greenland interior during the 2013 and 2014 seasons to measure accumulation rates over the majority of

the dry and upper percolation zones.

Regional climate models and reanalysis products provide spatially and temporally comprehensive estimates of accumulation at ice-sheet scales. The magnitude of mesoscale model uncertainty can be as large as the natural variability, or larger in areas with sparse *in situ* measurements like ice cores, potentially obscuring

climate fluctuations with random error (Burgess et al., 2010; Box et al., 2006). A 2013 study (Vernon et al.) determined that 1960-2008 climate model SMBs differ by as much as 130 Gt a$^{-1}$ across the ice sheet, with especially large differences in the southern (80.1 Gt a$^{-1}$) and northwestern (40.4 Gt a$^{-1}$) drainage basins. Many of the variables included in these models are validated with snow pits and ice cores, such as the
1997-1998 Program for Regional Climate Assessment (PARCA) ice core campaign (Mosley-Thompson et al., 2001), which predate the recent period of accelerated surface melting in Greenland (McGrath et al., 2013). We compare our IceBridge accumulation data with outputs from three RCMs to pinpoint their areas of highest uncertainty: (1) the Pennsylvania State University – National Center for Atmospheric Research Fifth-Generation Mesoscale Model (Polar MM5; Burgess et al., 2010), (2) the Regional Atmospheric
Climate MOdel (RACMO2.3; Noel et al., 2016), and (3) the Modèle Atmosphérique Régional (MARv3.5.2; Fettweis et al., 2016). We also compare accumulation results with a gridded land-ice accumulation dataset (Box et al., 2013; hereafter "Box13") and a krigged ice core accumulation record (Bales et al., 2009 hereafter "Bales09"). We further use principal component and correlation analyses to evaluate the dominant climate forcing mechanisms driving regional GrIS precipitation trends.

**2. Methods**

    **2.1. Accumulation Radar**

We calculate a spatially continuous record of accumulation along 17,730 km of NASA Operation IceBridge Accumulation Radar flights (hereafter "IceBridge accumulation"). Operation IceBridge was designed to bridge the gap in polar observations between the Ice, Cloud, and Land Elevation Satellite (ICESat; 2003-
2009) and ICESat-2, which is scheduled to launch in 2017. Laser altimeters, 4-5 different frequency radars, a gravimeter, and a magnetometer are mounted on NASA's P-3B and DC-8 airplanes, which conduct airborne surveys in both the Arctic and Antarctic each spring.

The IceBridge Accumulation Radar captures a continuous electromagnetic profile of the top few hundred
meters of the ice sheet, displaying distinct internal reflecting horizons (IRHs) that can be traced for hundreds of kilometers (Leuschen et al., 2011). The Accumulation Radar operates in the 600-900 MHz range and has an average vertical resolution of 0.28 m in snow/firn, which is fine enough to resolve IRHs that have been shown to represent isochrones (Medley et al., 2013; Rodriguez-Morales et al., 2014; Spikes et al., 2004; Hawley et al., 2014). The average distance between radar traces is 16 m, which we then
average over 10 adjacent traces to increase the signal-to-noise ratio. The position of each trace is known from differential GPS receivers mounted on the aircraft. We do not perform any time variable gain or additional filtering on the IceBridge accumulation data. Depending on signal attenuation within the

snowpack, IRHs can be traced to a depth of 50-150 m and provide accumulation records over the past 100-300 years (Figure 1). For areas with high attenuation (i.e. shallow penetration of the radar signal), such as those at relatively lower elevations (e.g. below ~2500 m), we calculate accumulation results for 1921-2014. Where the signal is less attenuated higher on the ice sheet, we calculate accumulation over the 1712-2014 time period (see Figure 2).

### 2.2. Depth-age scales and density profiles

To calculate accumulation rates using ice penetrating radar, one must know the amount of snow mass between IRHs and their relative ages. The mass between IRHs is a function of the depth-age scale, travel time-depth conversion rate, and firn or ice density. We obtain both the density profile and depth-age scale from two dated ice cores collected at Summit Station (Mary Albert, personal communication, 2015; Cole-Dai et al., 2009). These ice core sites are 3 and 7 km from the closest IceBridge radar trace, and we assume similar accumulation rates across this small distance. We correct for the 7-year difference between ice core collection and IceBridge radar flights by extrapolating the depth-age curve.

We calibrate a Herron-Langway (1980) depth-density model at Summit using data from both ice cores, then use the calibrated model parameters to estimate density profiles elsewhere in our study region. Input parameters for this model include satellite derived mean annual temperature (Hall et al., 2012), modelled accumulation (Burgess et al., 2010), and an estimate of surface snow density from field measurements along ground traverses, shallow firn cores, and MAR model output. Since we are using the density profile to calculate accumulation based, in part, on modelled accumulation, the result could be seen to be circular. However, our results are largely insensitive to changes in this modelled accumulation input because accumulation estimates are minimally affected by input variations to the Herron-Langway model. For example, adjusting input accumulation and surface density by ±5% results in <1% change in the calculated accumulation rates.

### 2.3. Travel-time to depth conversion

We convert the radar travel time to depth by iteratively multiplying the velocity of the electromagnetic wave by the signal's travel time to each IRH. The electromagnetic speed of the radar wave, $v$ (m s$^{-1}$), is calculated from the dielectric permittivity, $\varepsilon_r$ (dimensionless), and the speed of light in a vacuum, $c$ ($3\times10^8$ m s$^{-1}$), from

$$v = \frac{c}{\sqrt{\epsilon_r}}$$ **(Equation 1).**

In turn, the dielectric permittivity is calculated from the density, $\rho$ (g cm$^{-3}$), of snow and ice at depth for each radar trace (following Kovacs et al., 1995) by

$$\epsilon_r = (1.0 + 0.845 * \rho)^2 \quad \text{(Equation 2).}$$

The snow surface reflection is readily identified in each radar profile from the large signal amplitude. We then calculate the depth for each subsequent radar sample in the profile using the radar travel time and velocity profile from Eqn. 1 and 2, following Hawley et al. (2014).

### 2.4. Internal reflecting horizons

We manually select 19 clear, strong IRHs to consistently trace from Summit Station towards the NNW and SW along two main flight paths (April 5 and May 2, 2014, respectively; see Figure 1). When a layer appears to bifurcate due to changes in accumulation, we continue to trace the layer based on the trajectory of surrounding IRHs. Horizons are not traced in areas where the signal-to-noise ratio made them too difficult to discern.

Internal reflecting horizons for the other 23 flights in this study are traced from crossover locations with the two main flight paths. Wherever possible, we trace IRHs outwards from crossover locations along the two main flight paths to locations where those traced layers cross another flight path. Whenever we have accumulation differences at crossover locations larger than our accepted error, we review IRHs to determine which layers are incorrectly traced.

### 2.5. Accumulation calculations and uncertainty

Finally, we calculate snow accumulation using the ice core depth-age scales, modelled depth-density profiles, and traced IRHs. We calculate accumulation between each pair of adjacent IRHs for every radar trace along the flight lines. Spatial changes in accumulation are evident from varying vertical distances between IRHs along each flight line. Temporal changes in accumulation are evident from examining accumulation during different epochs at one radar trace. We calculate the water equivalent accumulation, $\dot{b}$ (m w.e. a$^{-1}$), between adjacent IRHs from the depth, $z$ (m) and age, $t$ (year), of each layer, the mean density, $\rho$ (kg m$^{-3}$), of each layer, and the density of water, $\rho_w$ (1000 kg m$^{-3}$):

$$\dot{b} = \frac{1}{t_2 - t_1} \int_{z_1}^{z_2} \frac{\rho(z)}{\rho_w} \partial z \quad \text{(Equation 3).}$$

We do not correct for ice flow due to advection of the ice sheet since nearly all of the radar traces occur in areas with surface velocities < 50 m a$^{-1}$. The only areas with higher velocities are across the Northeast Greenland Ice Stream and a high velocity region in the southwest. Velocities in these areas are ~60-100 m

a$^{-1}$ over the time domain of this study and do not significantly affect accumulation results. However, we do correct for layer thinning using a Nye (1963) model. For each radar trace, the thinning factor, $\lambda(z)$, is calculated from the average accumulation, $\dot{b}$ (m w.e. a$^{-1}$) of each epoch, average age of the epoch, $a$ (year), and water equivalent thickness of the GrIS, $H$ (m), from Morlighem et al. (2014):

5    $\lambda(z) = e^{-\frac{\dot{b}}{H}a}$        (**Equation 4**).

Uncertainty in accumulation can arise from independent errors in tracing IRHs, errors from incorrectly dating the ice core, and/or errors in the densities used for converting from separation distance to water equivalent accumulation.

To reduce tracing errors, two authors separately retraced each IRH along the two main flights paths four times each. Close inspection of the IRHs reveals that the peaks defining IRHs are within ±2 radar samples (within ± 0.557 m), and incorrectly jumping to the next layer would result in an error of at most ±5 samples (at most ± 1.39 m). Our average epoch between IRHs is 16.7 years, which corresponds to a maximum error 15    of ~±0.083 m a$^{-1}$.

We take uncertainty in dating the Summit ice cores to be ±1% for the top 100 years, ±2% for 100-200 years ago, and ±3% for 200-300 years ago. The oldest isochrones traced in this study are dated to 1712, which suggests a maximum error of 3% using a 2007 Summit Station ice core. At the lowest accumulation 20    locations, the smallest distance between layers is 0.26 m w.e. over an epoch of 5.18 years. This gives an uncertainty in accumulation due to dating of ~±0.03 m w.e. a$^{-1}$.

The error associated with measuring density using similar techniques has been estimated to be 1.4% (Karlöf et al., 2005). However, following Hawley et al. (2014) we conservatively assume that our 25    measurements have an error of up to twice this large, corresponding to a maximum accumulation error of ±0.014 m w.e. a$^{-1}$.

The three error sources are all random, non-systematic, and thus can be assumed to be non-additive (following Hawley et al., 2014). Over the extent of the dataset we can assume that the errors are not 30    correlated, thus we estimate accumulation uncertainty from all sources at ±0.127 m w.e. a$^{-1}$ for any single epoch. Due to the random and non-systematic nature of these errors, we can assume that they are unlikely to contribute to a regional or temporal accumulation bias. To calculate uncertainty for accumulation

averaged over multiple epochs, we divide our uncertainty by the square root of the number of traced layers at that location.

### 2.6. Model comparison

We compare our IceBridge accumulation results with annual outputs from Polar MM5 (1958-2008; Burgess et al., 2010), MARv3.5.2 (1948-2015; Fettweis et al., 2016), RACMO2.3 (1958-2015; Noel et al., 2016), and Box13 (1840-1999; Box et al., 2013). Grid cell sizes for these model outputs are 24 km, 5 km, 1 km, and 5 km, respectively. Since accumulation can be bilinearly interpolated over the distance of these grid cells without significant loss of detail (Box and Rinke, 2003), we choose to compare IceBridge accumulation with bilinearly interpolated model grid output to compare accumulation at corresponding spatial locations.

The Box13 dataset is corrected using a correction multiplier grid, which is estimated using a triangular irregular network interpolation of the ratio between 1961-1990 average Box13 ice core accumulation rates and RACMO2.1 output. The multipliers have respective minimum and maximum values of 0.605 and 1.891. We assume that the calibration coefficients are stationary in both time and space, since Fettweis et al., (2016) show that MAR accumulation reconstructions are similar to those from Box13 after 1930.

Additionally, we compare our IceBridge accumulation with an accumulation map kriged from 295 snow pits and ice cores and 20 coastal weather stations (Bales et al., 2009). While this map estimates accumulation over the time domain of the oldest ice cores, we choose to compare IceBridge accumulation with the highest accuracy accumulation estimates from 1950-2000, which include weather stations and recent ice cores.

## 3. Results and discussion

### 3.1. IceBridge accumulation rates

IceBridge accumulation patterns are consistent with observed large-scale spatial patterns from ice cores and snow pits (Bales et al., 2009), with high accumulation rates in the southeast and southwest and lower accumulation rates in the northeast and at higher elevations of the ice sheet interior (Figure 3). The number of traceable layers is highest towards the interior of the ice sheet and lowest in warmer areas towards the coast and in the south, where enhanced surface melt attenuates the radar signal and reduces the density gradients that produce IRHs (Figure 2).

We assess the internal consistency of IceBridge accumulation by comparing the accumulation at 87 locations where IceBridge flight paths cross one another (hereafter "crossover points"). Differences at crossover points are most likely due to errors in layer picking where isochrones become difficult to detect or distinguish. There are no spatial or temporal patterns in accumulation differences at crossover points over the dataset. Moreover, the differences are normally distributed with a mean of $0.017 \pm 0.022$ m w.e. a$^{-1}$ (n = 1241), and all but five crossover point accumulation differences fall within our calculated uncertainty of 0.127 m w.e. a$^{-1}$ (Figure 4).

### 3.2. Validation with in-situ measurements

Accumulation rates derived from ice cores collected at Camp Century, D3, and D4 (see Figure 2 for locations) correspond closely with our IceBridge accumulation rates, matching their long-term mean and tracking their decadal variability (Figure 5). Additionally, we compare IceBridge accumulation rates and trends to the NASA-U, NEEM, D5, B26, B29, NGRIP, and PARCA ice cores over corresponding temporal domains (Table 1). IceBridge accumulation rates and accumulation trends are statistically indistinguishable from each of these cores at a $p < 0.05$ confidence level using a Student's t-test.

In Figure 6 we compare IceBridge accumulation to snowpit measurements at station T-31 on the Expédition Glaciologique Internationale au Groenland (EGIG) traverse (Fischer et al., 1995; Hurbertus Fischer, personal communication., 2015), and to accumulation rates calculated at this location from the Airborne SAR/Interferometric Radar Altimeter System (ASIRAS; Overly et al., 2016) (see Figure 2 for location). IceBridge accumulation rates are statistically indistinguishable ($p < 0.05$) from both snowpit measurements and ASIRAS accumulation results (Figure 6).

### 3.3. Comparison to modelled accumulation

We compare IceBridge accumulation to RCM accumulation results along the length of each flight. IceBridge accumulation is averaged over 1957-2014 to compare with averaged Polar MM5 (1958-2008), MAR (1948-2015), RACMO2 (1958-2015), and Box13 (1840-1999). An example of this comparison along a single flight (B-B'-B'' in Figure 2) is shown in Figure 7. Differences between the IceBridge accumulation and RCM output are spatially heterogeneous along the flight path, varying in both location and magnitude. Averaged over the entire length of the flight, Polar MM5 underestimates accumulation by $0.001 \pm 0.010$ m w.e. a$^{-1}$, MAR underestimates by $0.006 \pm 0.012$ m w.e. a$^{-1}$, RACMO2 overestimates by $0.008 \pm 0.011$ m w.e. a$^{-1}$, Box13 underestimates by $0.028$ and $\pm 0.017$ m w.e. a$^{-1}$, and Bales09 overestimates by $0.007 \pm$

0.014 m w.e. a$^{-1}$. In addition, the high spatial resolution of our dataset shows significant accumulation variability not captured in model outputs.

The model output and IceBridge accumulation time domains do not match identically, but these minor differences do not significantly affect our results. The largest time domain discrepancy is with the Polar MM5 comparison, where model output is averaged from 1958-2008 and IceBridge accumulation is averaged from 1957-2014. The top panel of Figure 7 shows Polar MM5 output averaged from 1958-2008 compared to IceBridge accumulation averaged from 1957-2004. The difference between IceBridge averaged over 1957-2014 and IceBridge averaged over 1957-2004 along this flight is 0.00096 ± 0.0021 m w.e. a$^{-1}$, well within calculated error.

Next, we compute the magnitude and percent differences between RCM output and IceBridge accumulation over the entire domain of this dataset. Averaged over all 25 flights, the RMS difference between the models and IceBridge accumulation is 0.036 ± 0.022 m w.e. a$^{-1}$ for Polar MM5, 0.023 ± 0.019 m w.e. a$^{-1}$ for RACMO2, 0.043 ± 0.029 m w.e. a$^{-1}$ for MAR, and 0.033 ± 0.026 m w.e. a$^{-1}$ for Box13. These average RMS errors are remarkably small, but Figure 8 shows considerably larger model-specific regional differences between IceBridge accumulation and RCM output. It is worth noting these differences are a significant improvement from previous versions of the regional climate model output. For example, the RMS difference between model and IceBridge accumulation for MARv3.2 (~2013) is 0.064 ± 0.033 m w.e. a$^{-1}$ and for RACMO2.1 (~2014) is 0.043 ± 0.018 m w.e. a$^{-1}$. These results highlight the importance of updated RCMs and additional in situ data to continually validate model results for improved Greenland SMB calculations.

We divide the Greenland ice Sheet into six major drainage basins (see Figure 8) following Vernon et al. (2013) to evaluate and discuss the spatial differences between model and IceBridge accumulation. Table 2 shows both percent and magnitude differences between the models and 1957-2014 averaged IceBridge accumulation in each of the six drainage basins. Statistically significant differences ($p < 0.05$) are highlighted in bold.

Averaged across basin A, the northern basin with generally low accumulation rates, there are no statistically significant differences between IceBridge accumulation and any of the RCMs used in this study. Although the models disagree with each other in this basin, as suggested by Vernon et al. (2013), the differences from the IceBridge accumulation are neither large nor statistically significant. Basin B in the northeast has some of the largest differences between models and IceBridge accumulation. Averaged across

all 815 points in basin B, MAR and Box13 underestimate by 18.68 ± 9.29% and 17.29 ± 6.30%, respectively. Basin C in the east also has significant differences between model and IceBridge accumulation; Polar MM5 underestimates by 9.45 ± 3.80% and MAR overestimates by an average of 20.89 ± 6.75%, although it overestimates by as much as 44.7 ± 7.8% in several locations (Figure 8e-f).

Basin D in the southeast is poorly covered by our data, but we find that MAR significantly overestimates accumulation by an average of 23.31 ± 5.36%. Koenig et al. (2016) similarly found that MAR overestimates accumulation in the SE region for the years 2009-2011 in comparison to IceBridge snow radar accumulation rates. Averaged across basin E, there are no statistically significant differences between 10 IceBridge accumulation and any of the RCMs used in this study. Likewise, Vernon et al. (2013) finds little difference in basin E between the climate models used in that study. On the other hand, Polar MM5 underestimates accumulation in basin F, with a statistically significant of 11.32 ± 5.28%. Figure 8 shows that the differences are particularly large near Camp Century (see Figure 2 for reference), where Polar MM5 underestimates by 16.15 ± 3.75% and MAR overestimates by 22.98 ± 6.79%.

In summary, the RCMs do an excellent job of calculating accumulation averaged over basins A and E, but there are large differences between model and IceBridge accumulation in basins B and C. We note that RACMO2.3 does not significantly differ from IceBridge accumulation in any of the basins. Areas where RCM and IceBridge accumulation differ the most are concurrent with areas without many *in situ* 20 measurements (e.g. in the southeast), and where ice cores were collected several decades ago (e.g. NASA-U, Camp Century). Additional field measurements would be beneficial to validate both our IceBridge accumulation and RCMs in these data-poor regions.

Averaged across all 25 flights, the Bales09 accumulation model kriged from ice core and snow pit 25 measurements differs from averaged 1957-2014 IceBridge accumulation by 0.033 ± 0.023 m w.e. a$^{-1}$ (Figure 8i-j). There are no statistically significant differences between Bales09 and IceBridge accumulation in any of the six drainage basins (Table 2), although differences are also largest in areas with sparse *in situ* measurements.

30 Basins B, E and F have sufficient data coverage to extrapolate over these basins' spatial domain to estimate the model uncertainty of their SMB estimates. We obtain total model uncertainty (in GT a$^{-1}$) by multiplying the percent difference in Table 2 by the annual regional SMB in each basin over 1961-1990 (Table 3 from Vernon et al., 2013). For basins B, E, and F, MAR differs by a combined total of -19.63 to 10.17 Gt a$^{-1}$, RACMO2 differs between -13.97 to 10.77 GT a$^{-1}$, and Polar MM5 underestimates by 6.84 to 30.78 Gt a$^{-1}$.

Given a modelled GrIS SMB of $363 \pm 89$ GT a$^{-1}$ (Vernon et al., 2013), the uncertainties in these three basins represent a total SMB difference of -5.41% to 2.80% (MAR), -3.84% to 2.96% (RACMO2) or an underestimation of 1.88% to 8.48% (Polar MM5). Today, it would take 360 GT of ice mass loss to raise global sea level by 1 mm. Thus, the combined MAR SMB underestimation from basins B, E, and F could represent up to 0.054 mm a$^{-1}$ of less sea level rise than previously imagined from the GrIS.

### 3.4. Comparison with Karlsson et al. (2016)

A study by Karlsson et al. (2016; hereafter Karlsson16) uses a very different method to calculate accumulation from IceBridge Accumulation Radar data near NEEM and NGRIP. We compare data from their study, representing flight lines in 2011 and 2012, to a repeat flight during the 2014 IceBridge season analyzed using our method. In Figure 9, the 1921-2014 accumulation rates (this study) are plotted against 1911-2011 Karlsson16 accumulation rates and the RCMs used for comparison in this study. On average along the 350 km flight line, the accumulation rates calculated in this study are $0.002 \pm 0.005$ m w.e. a$^{-1}$ higher than in Karlsson16, well within calculated error, and in better agreement than either dataset with the RCMs. Our accumulation values agree better with Karlsson16 from 150 km along the transect to NGRIP (underestimate of $0.002 \pm 0.002$ m w.e. a$^{-1}$) than they do along the first half of the transect (overestimate of $0.007 \pm 0.004$ m w.e a$^{-1}$). The average 1817-1921 measurements (this study) are 0.01 m w.e. a$^{-1}$ higher than the 1811-1911 Karlsson16 values, and the 1712-1811 measurements (this study) are 0.0081 m w.e. a$^{-1}$ higher than the 1711-1811 Karlsson16 values. Thus, our results are nearly identical with Karlsson16 over the time domain of this study, despite the two studies using different methods to calculate accumulation, analyzing different IceBridge flights from different years, and tracing IRHs from different ice cores.

### 3.5. IceBridge accumulation temporal trends

We can analyze spatiotemporal trends in snow accumulation using our IceBridge accumulation record spanning 17,700 km of flight paths over the past 300 years. We perform an empirical orthogonal function (EOF) analysis on the dataset to evaluate temporal changes in accumulation and assess potential atmospheric forcing mechanisms (Figure 10). We limit our EOF analysis to 1889-2014 to capture the maximum spatial variability since layers older than 1889 are difficult to trace in the southern region (see Figure 2). We find that EOF1 and EOF2 represent most of the variance within the dataset, explaining 33% and 19% of the variance, respectively.

The EOF1 time series has a statistically significant positive correlation with the 1899-2014 annually averaged Atlantic Multidecadal Oscillation (AMO) index ($r = 0.60$, $p < 0.04$), the wintertime (DJF) AMO

(r = 0.55, p < .05), and the springtime (MAM) AMO (r = 0.56, p < 0.05). These correlations indicate an association between the AMO and Greenland precipitation, although due to collinearity, any physical relation could partly be acting through NAO changes. Figure 10a indicates that while the majority of the ice sheet has a positive correlation with the AMO, Camp Century and NW Greenland have a weak negative correlation. This same pattern is produced by a Pearson correlation between the annual AMO index and IceBridge accumulation (Figure 11c), although the negative correlations in NW Greenland are not statistically significant. This pattern is consistent with the results of Chylek et al. (2012), who found a dominant AMO cycle of 20 years in several ice cores collected from southern and central Greenland, but did not observe an AMO signal in NW Greenland.. Mernild et al. (2014) similarly found a significant positive relationship between the AMO and a composite Greenland ice core precipitation record from 1890-2000. The positive GrIS precipitation correlation with the AMO may be due to warmer North Atlantic and Greenland temperatures during AMO positive conditions, leading to higher absolute humidity from the Clausius-Clapeyron relationship (Held and Soden, 2006). It is also possible that this correlation may be due to associated storm-track changes from warmer North Atlantic and Greenland temperatures (e.g. Hanna et al., 2013b, 2016).

The EOF2 time series is significantly correlated with the wintertime (DJF) North Atlantic Oscillation (NAO), with r = 0.62 (p < 0.03) for the Hurrell (1995) principal component-based NAO index and r = 0.60 (p < 0.04) for the Jones et al. (1997) station-based NAO index Figure 10. Negative correlations in the northern and western regions of our study area are indicative of greater precipitation during NAO negative conditions, when the Icelandic Low and Azores High pressure centers weaken and there is enhanced southerly flow of warm, moist air masses into Baffin Bay (Hurrell, 1995). Banta and McConnell (2007) and Mosley-Thompson et al. (2005) likewise document negative correlations between the NAO and ice core accumulation in central western and northwestern Greenland (e.g. NASA-U, D3, D4; see Figure 2 for locations). Mernild et al. (2014) also find a significant influence of the NAO on their composite coastal Greenland precipitation record, and both Wong et al. (2015) and Osterberg et al., (2015) find significant negative correlations between the NAO and precipitation and temperature, respectively, at Thule in northwest Greenland. The EOF2 loading is also weak in the region of Summit (Figure 10), consistent with the findings of Mosley-Thompson et al. (2005) and Banta and McConnell (2007). Interestingly, the EOF2 loading pattern reflects a generally southeast-northwest dipole in accumulation response to the NAO, which differs from the dominantly east-west dipole response to the NAO in reanalysis data (not shown). Varimax rotation of the IceBridge EOF2 did not significantly change the orientation of the dipole. This dipole pattern is reproduced in a direct Pearson correlation between the wintertime NAO index and IceBridge accumulation (Figure 11a).

Although it does not appear through our EOF analysis, there are significant positive correlations between the summertime GBI and IceBridge accumulation (Figure 11b), indicating positive GrIS accumulation anomalies during summers with overall enhanced blocking. While this may seem counter-intuitive, this relationship is driven by enhanced meridional flow and moisture advection into Greenland under the weak zonal flow associated with GBI positive (generally NAO negative) conditions (Hanna et al., 2016). Hanna et al. (2016), in a study based on reanalysis data, similarly find enhanced precipitation in central-northern Greenland associated with positive GBI summers (their Figure 6g). They also show negative precipitation anomalies in southeast Greenland during positive GBI summers, but our IceBridge data coverage in that region is too poor to confidently evaluate GBI relationships.

If our hypothesis is correct that a positive AMO index (anomalously warm North Atlantic sea-surface temperatures) contributes to anomalously high GrIS accumulation, then the future behavior of the AMO may have a significant impact on the rate of GrIS mass loss. Hanna et al. (2013b) found that positive AMO summers were associated with enhanced GrIS surface melting, indicating that the AMO impacts both the mass input and mass loss portions of Greenland SMB. The highest quality climate observations, reanalysis data and RCM output exist for the 1979-present interval, during which the AMO progressed from a negative phase (in the 1980s) to a positive phase (in the 2000's), with a rapid AMO warming transition in the 1990s (Figure 10c). Paleoclimate records show evidence that the AMO was a persistent sea surface temperature (SST) mode throughout the late Holocene with a periodicity of 20-70 years (Chylek et al., 2012; Knudsen et al., 2011), and thus would be expected to continue into the future. We therefore encourage modeling efforts to evaluate the GrIS mass balance implications of a future return towards AMO negative conditions during a continued increase in radiative forcing from anthropogenic greenhouse gases.

## 4. Conclusions

We have developed a new dataset of accumulation rates over the interior of the Greenland ice sheet spanning the past 100-300 years based on 17,730 km of Operation IceBridge airborne Accumulation Radar data. This accumulation record is internally consistent across the dataset and is validated by *in situ* field measurements, several ice cores, and other radar-derived accumulation measurements.

Overall, the Polar MM5, MAR, and RACMO2 Regional Climate Models, as well as Box13 and Bales09, accurately capture large spatial patterns in accumulation over the GrIS, but show significant differences from IceBridge accumulation on a regional basis. For example, in the southeast MAR overestimates

accumulation by an average of 20.89 ± 6.75% and as much as 44.7 ± 7.8% in several locations. These RCM differences could lead to regional Greenland mass balance errors ranging between an underestimate of 30.78 GT $a^{-1}$ and an overestimate of 10.77 GT $a^{-1}$ for the northwest, west, and northeastern drainage basins. These combined regional uncertainties represent up to 8.48% of the total GrIS SMB, and an equivalent of

up to 0.054 mm $a^{-1}$ of less sea level rise than predicted.

Empirical orthogonal function analysis indicates that the first and second principal components explain 33% and 19% of the variance and correlate with the AMO and NAO, respectively. These results are consistent with previous ice core and weather station analyses demonstrating the importance of these North

Atlantic climate models on Greenland SMB. We recommend that future modelling efforts evaluate the effects of a future return to AMO negative conditions on GrIS surface mass balance as greenhouse gas concentrations continue to rise.

Our largest accumulation uncertainties align with regions that disagree most strongly with climate models.

Thus, future research should be aimed at collecting additional *in situ* measurements in areas with large disagreement between climate models, particularly in the southeast.

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

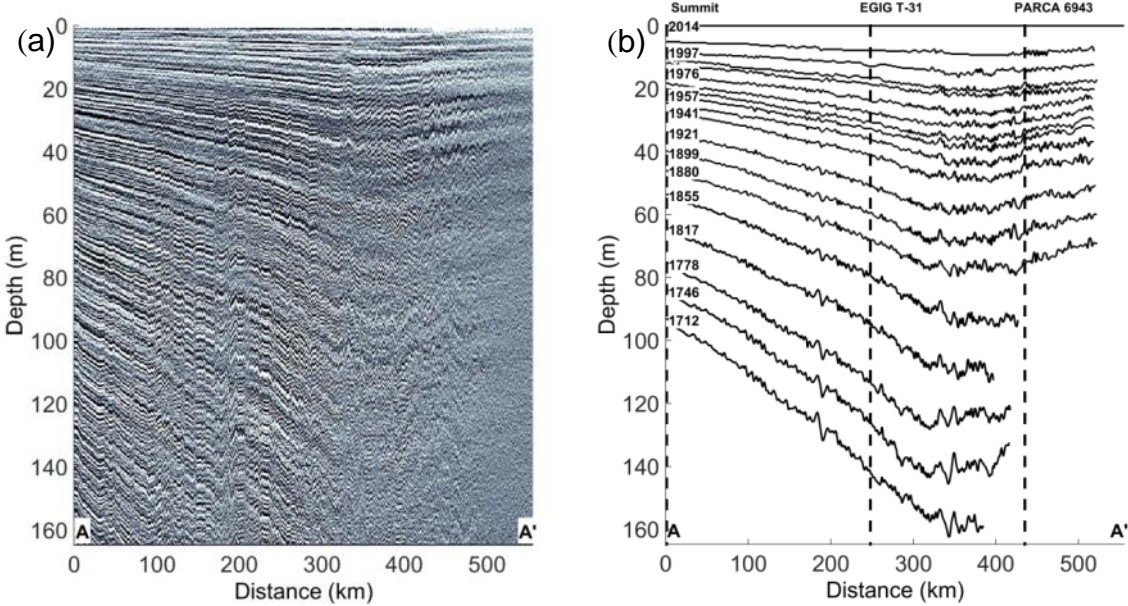

**Figure 1: a) Radargram showing flight A-A' (see Figure 2 for location). b) Nineteen traced internal reflecting horizons from two dated ice cores at Summit Station through EGIG T-31 and the PARCA 6943 ice core.**

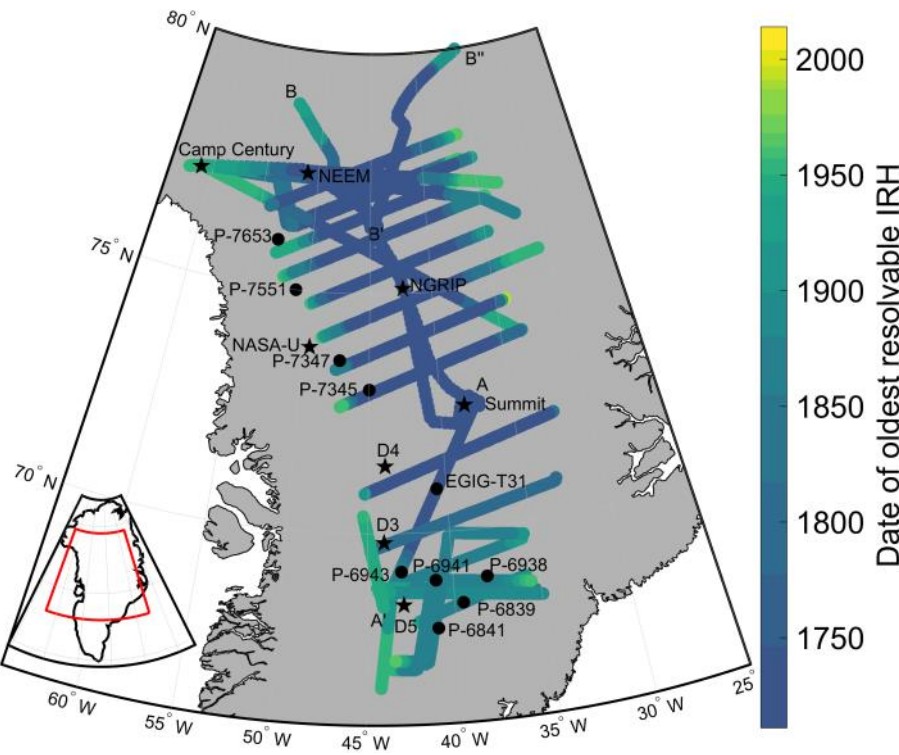

**Figure 2: Date of oldest resolvable Internal Reflection Horizon (IRH) along 25 IceBridge Accumulation Radar flights totaling 17,730 km. Locations are shown for A-A' (Figure 1) and B'-B'-B'' (Figure 7) as well as EGIG-T31 and D3, D4, D5, NEEM, NGRIP, NASA-U, Camp Century, and PARCA ice cores (see Figure 5 and Table 1).**

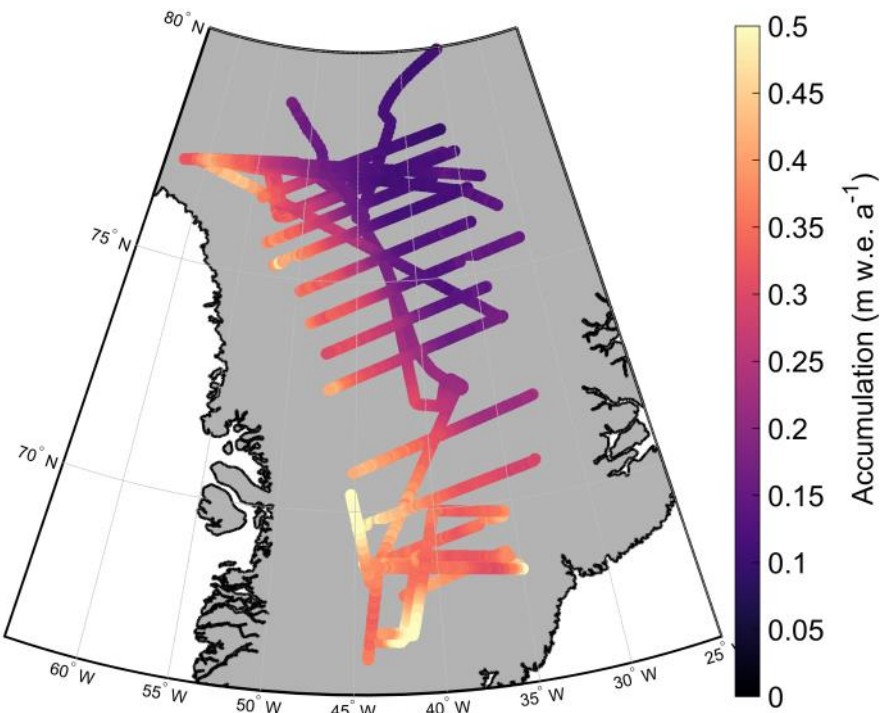

**Figure 3: Average accumulation over the temporal domain of each radar trace calculated from IceBridge Accumulation Radar over all 25 flights. IceBridge accumulation matches large-scale accumulation patterns from ice cores and snow pits from Bales et al. (2009).**

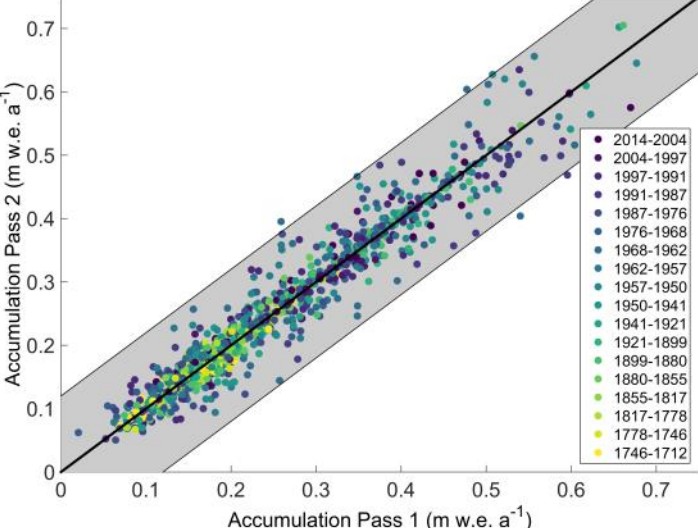

Figure 4: Comparison of IceBridge accumulation rates determined at 87 crossover locations for each epoch, totaling 1241 measurements. There are no temporal or spatial patterns in crossover location accumulation differences. Shaded region is the calculated uncertainty of ± 0.127 m w.e. a⁻¹.

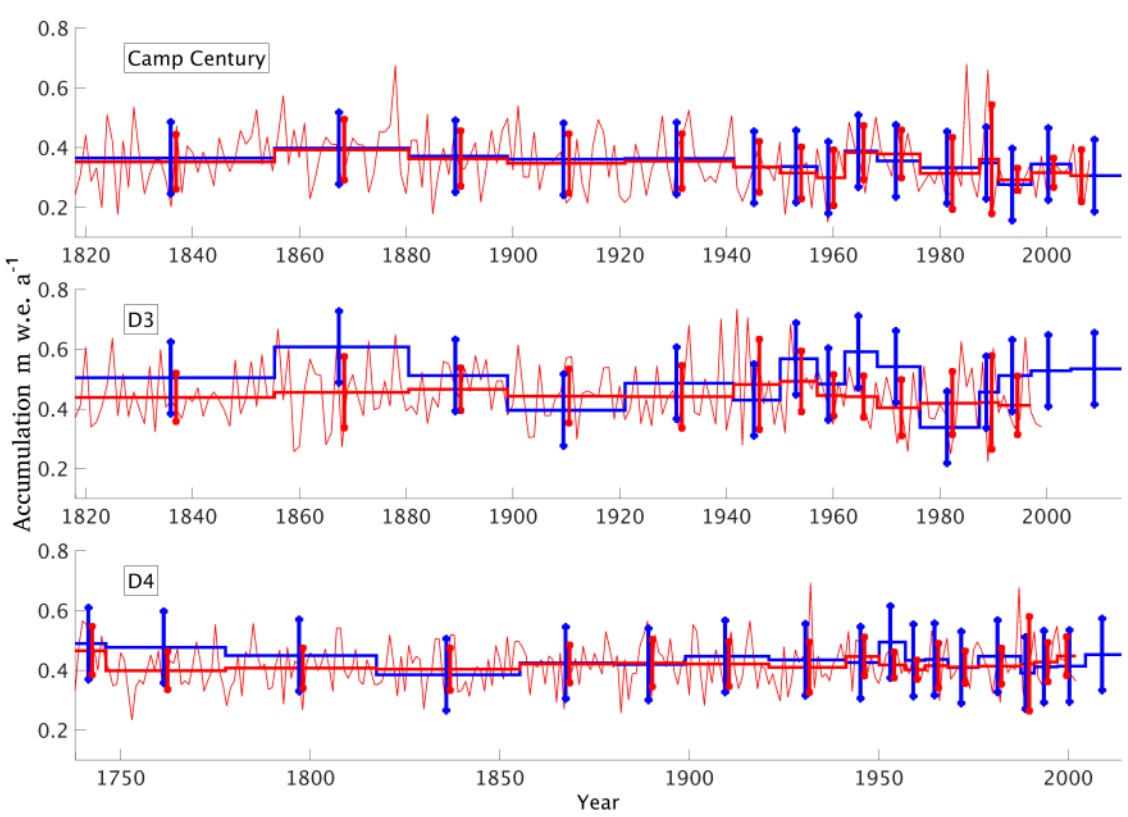

Figure 5: IceBridge accumulation (blue) with uncertainty (blue circles) compared with Camp Century, D3, and D4 (see Figure 2 for locations) ice core annual accumulation (thin red lines) and ice core accumulation averaged over corresponding epochs (thick red lines). One standard deviation of ice core annual accumulation over each epoch is shown with a red square. Note the longer time scale for the D4 ice core. There is no statistically significant difference between IceBridge and ice core accumulation for any of these ice cores.

**Table 1: Averaged ice core accumulation compared with IceBridge accumulation averaged over the overlapping time domain of each ice core. Uncertainty figures represent one standard deviation of ice core accumulation and average IceBridge accumulation at the closest radar trace to each core, respectively. Trends and their standard deviation are reported for both ice core accumulation and nearest IceBridge accumulation.**

| Ice Core | Average Ice Core Accumulation (m w.e. a$^{-1}$) | Average IceBridge Accumulation (m w.e. a$^{-1}$) | Time period of comparison | Trend (core) (mm w.e. a$^{-2}$) | Trend (IB) (mm w.e. a$^{-2}$) |
|---|---|---|---|---|---|
| Nasa-U | $0.35 \pm 0.07$ | $0.36 \pm 0.07$ | 1921-1991 | $-0.09 \pm 0.38$ | $-0.38 \pm 0.95$ |
| NEEM | $0.19 \pm 0.04$ | $0.21 \pm 0.06$ | 1855-2004 | $-0.06 \pm 0.08$ | $-0.08 \pm 0.41$ |
| Camp Century | $0.35 \pm 0.10$ | $0.35 \pm 0.03$ | 1817-2004 | $-0.24 \pm 0.13$ | $-0.39 \pm 0.15$ |
| D3 | $0.45 \pm 0.10$ | $0.49 \pm 0.07$ | 1836-1999 | $-0.08 \pm 0.09$ | $-0.16 \pm 0.40$ |
| D4 | $0.42 \pm 0.07$ | $0.43 \pm 0.06$ | 1746-2002 | $0.11 \pm 0.06$ | $0.14 \pm 0.15$ |
| D5 | $0.38 \pm 0.07$ | $0.38 \pm 0.07$ | 1941-2002 | $0.12 \pm 0.51$ | $0.10 \pm 0.92$ |
| B26 | $0.18 \pm 0.03$ | $0.19 \pm 0.04$ | 1712-1991 | $-0.03 \pm 0.03$ | $0.01 \pm 0.12$ |
| B29 | $0.16 \pm 0.03$ | $0.18 \pm 0.04$ | 1712-1991 | $0.00 \pm 0.02$ | $-0.03 \pm 0.12$ |
| NGRIP | $0.19 \pm 0.03$ | $0.19 \pm 0.03$ | 1712-1997 | $-0.02 \pm 0.02$ | $-0.04 \pm 0.11$ |
| P-6839 | $0.39 \pm 0.15$ | $0.39 \pm 0.08$ | 1987-1997 | | |
| P-6841 | $0.48 \pm 0.16$ | $0.45 \pm 0.03$ | 1987-1997 | | |
| P-6938 | $0.36 \pm 0.07$ | $0.34 \pm 0.05$ | 1987-1997 | | |
| P-6941 | $0.40 \pm 0.10$ | $0.40 \pm 0.03$ | 1987-1997 | | |
| P-6943 | $0.39 \pm 0.10$ | $0.40 \pm 0.07$ | 1976-1997 | | |
| P-7345 | $0.28 \pm 0.07$ | $0.32 \pm 0.07$ | 1976-1997 | | |
| P-7347 | $0.29 \pm 0.09$ | $0.33 \pm 0.09$ | 1976-1997 | | |
| P-7551 | $0.32 \pm 0.09$ | $0.30 \pm 0.08$ | 1962-1997 | | |
| P-7653 | $0.35 \pm 0.09$ | $0.40 \pm 0.09$ | 1976-1997 | | |

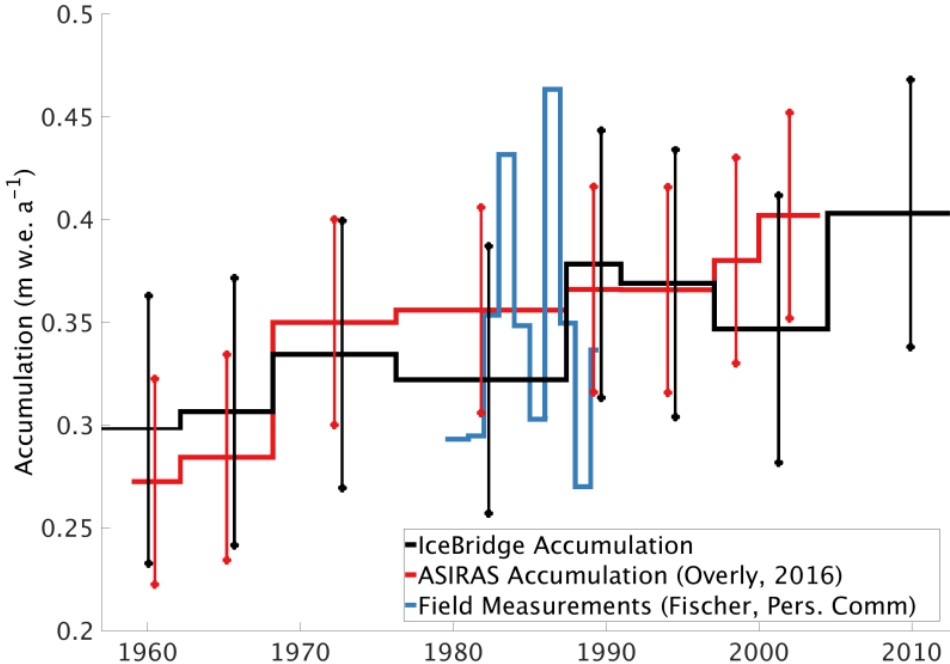

**Figure 6: IceBridge accumulation results at EGIG T-31 (see Figure 2 for location) from 1957-2014 are statistically indistinguishable from Airborne SAR/Interferometric Radar Altimeter System (ASIRAS) accumulation (Overly et al., 2016), and field measurements (H. Fischer, personal communication, 2015). Error bars are 1 standard deviation of ASIRAS accumulation over data points from that time period.**

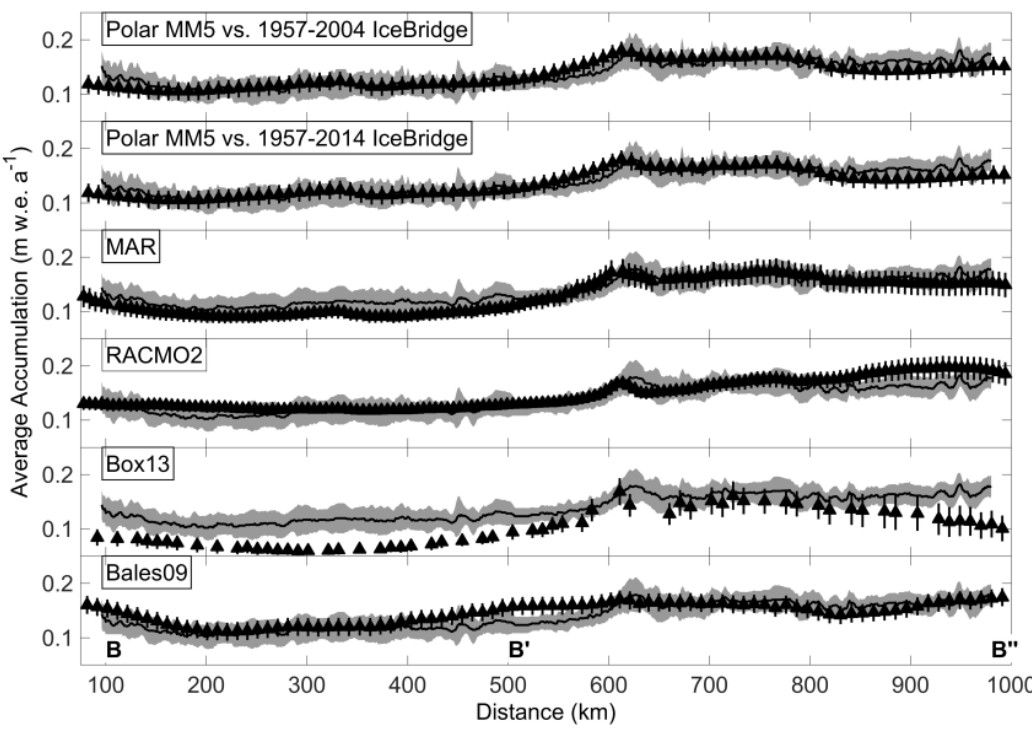

**Figure 7: Comparison of 1957-2004 averaged IceBridge accumulation (solid line) and uncertainty (shaded region) to averaged Polar MM5 (1958-2008; triangles) along a 977 km flight in northern Greenland. Location of flight shown as B-B'-B'' on Figure 2. Comparison of 1957-2014 averaged IceBridge accumulation to averaged Polar MM5 (1957-2008), MAR (1948-2015), RACMO2 (1958-2015), and Bales09 accumulation along the same flight. The difference between 1957-2004 and 1957-2014 IceBridge accumulation across this flight is insignificant.**

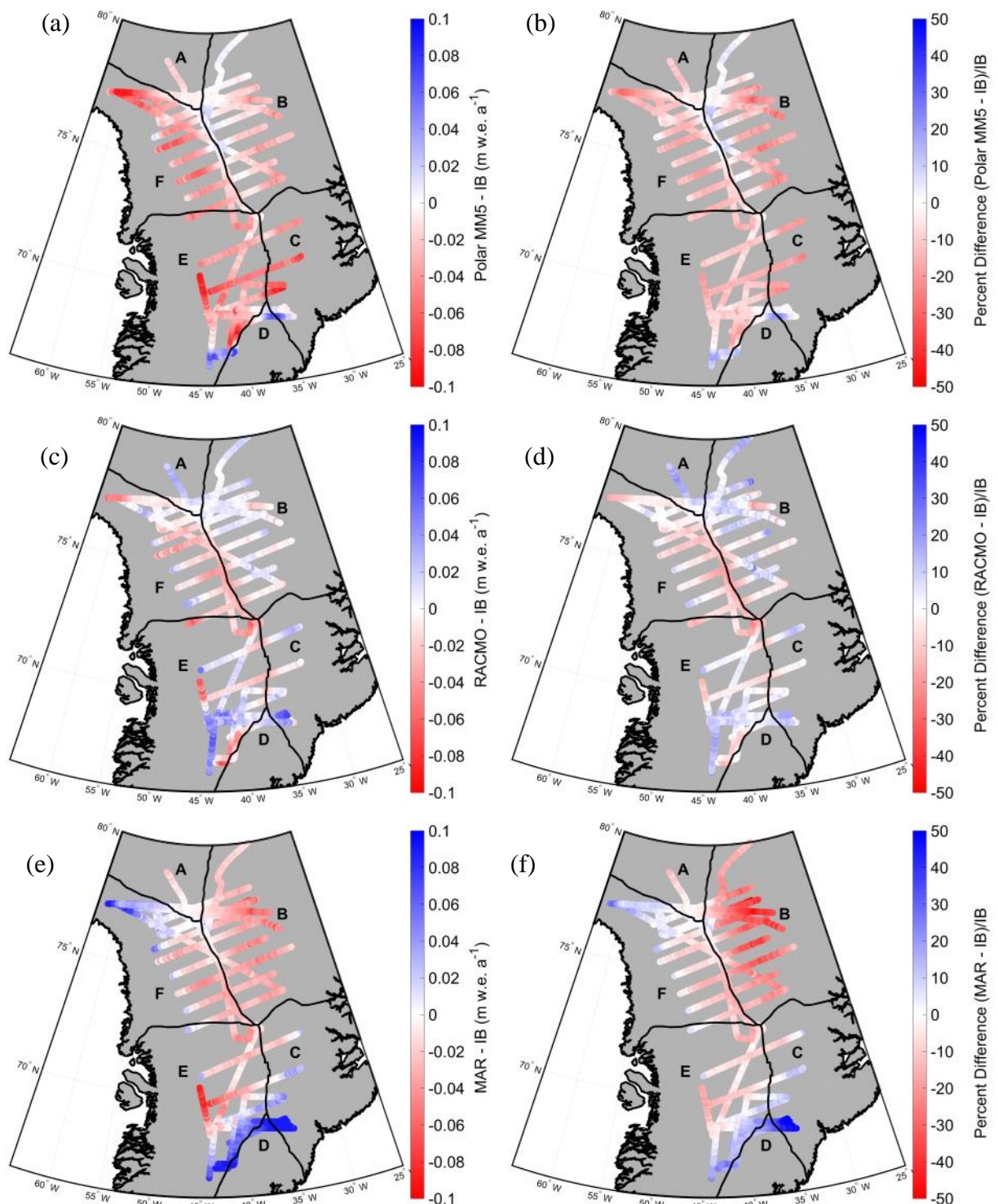

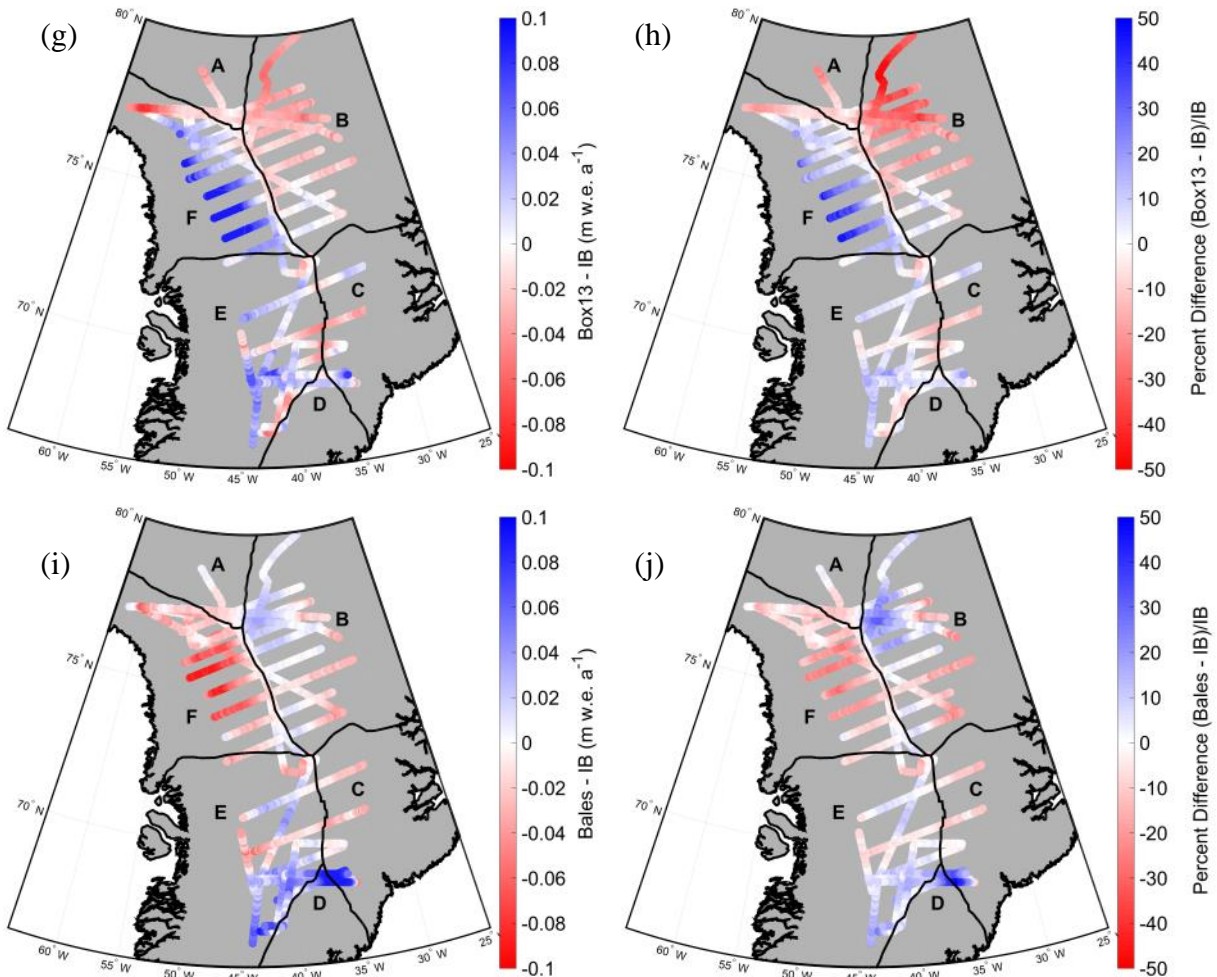

**Figure 8: Magnitude (left) and percent (right) differences between averaged 1957-2014 IceBridge accumulation and (a-b) Polar MM5, (c-d) RACMO2, (e-f) MAR, (g-h) Box13, and (i-j) Bales09 averaged accumulation. Also shown are six drainage basins of the Greenland Ice Sheet discussed in the text (c.f. Vernon et al., 2013).**

**Table 2: Percent and magnitude differences between average 1957-2014 IceBridge accumulation and average model accumulation in each of the six Greenland Ice Sheet drainage basins. Positive numbers indicate that the model overestimates accumulation in that basin. Plus minus figures represent one standard deviation. Statistically significant differences are indicated in bold.**

| | A (n = 135) | B (n = 815) | C (n = 234) | D (n = 102) | E (n = 1064) | F (n = 831) |
|---|---|---|---|---|---|---|
| Polar MM5 (%) | -2.73 ± 3.73 | -7.60 ± 8.00 | **-9.45 ± 3.80** | 4.33 ± 5.85 | -7.96 ± 4.73 | **-11.32 ± 5.28** |
| RACMO2 (%) | 6.22 ± 5.25 | 1.67 ± 6.65 | 4.73 ± 5.36 | -0.20 ± 2.44 | 1.97 ± 5.98 | -5.12 ± 5.15 |
| MAR (%) | 3.66 ± 4.22 | **-18.68 ± 9.29** | **20.89 ± 6.75** | **23.31 ± 5.36** | 3.39 ± 5.88 | 2.83 ± 6.96 |
| Box13 (%) | -6.85 ± 4.47 | **-17.29 ± 6.30** | -0.14 ± 5.28 | -0.35 ± 2.33 | 3.99 ± 5.97 | 5.36 ± 8.66 |
| Bales09 (%) | -4.28 ± 4.64 | 3.35 ± 9.26 | 5.62 ± 5.65 | 16.91 ± 9.03 | 4.76 ± 5.00 | -9.13 ± 5.24 |
| | | | | | | |
| Polar MM5 (m w.e. a$^{-1}$) | -0.005 ± 0.006 | -0.010 ± 0.011 | **-0.027 ± 0.012** | 0.015 ± 0.022 | -0.031 ± 0.020 | -0.036 ± 0.020 |
| RACMO2 (m w.e. a$^{-1}$) | 0.009 ± 0.009 | 0.001 ± 0.009 | 0.015 ± 0.017 | -0.002 ± 0.010 | 0.007 ± 0.022 | -0.014 ± 0.015 |
| MAR (m w.e. a$^{-1}$) | 0.006 ± 0.007 | **-0.024 ± 0.012** | **0.075 ± 0.026** | **0.085 ± 0.015** | 0.012 ± 0.023 | 0.011 ± 0.023 |
| Box13 (m w.e. a$^{-1}$) | -0.012 ± 0.007 | **-0.021 ± 0.008** | -0.001 ± 0.017 | -0.001 ± 0.009 | 0.014 ± 0.022 | 0.016 ± 0.029 |
| Bales09 (m w.e. a$^{-1}$) | -0.008 ± 0.008 | 0.003 ± 0.012 | 0.022 ± 0.017 | 0.058 ± 0.028 | 0.016 ± 0.019 | -0.028 ± 0.017 |

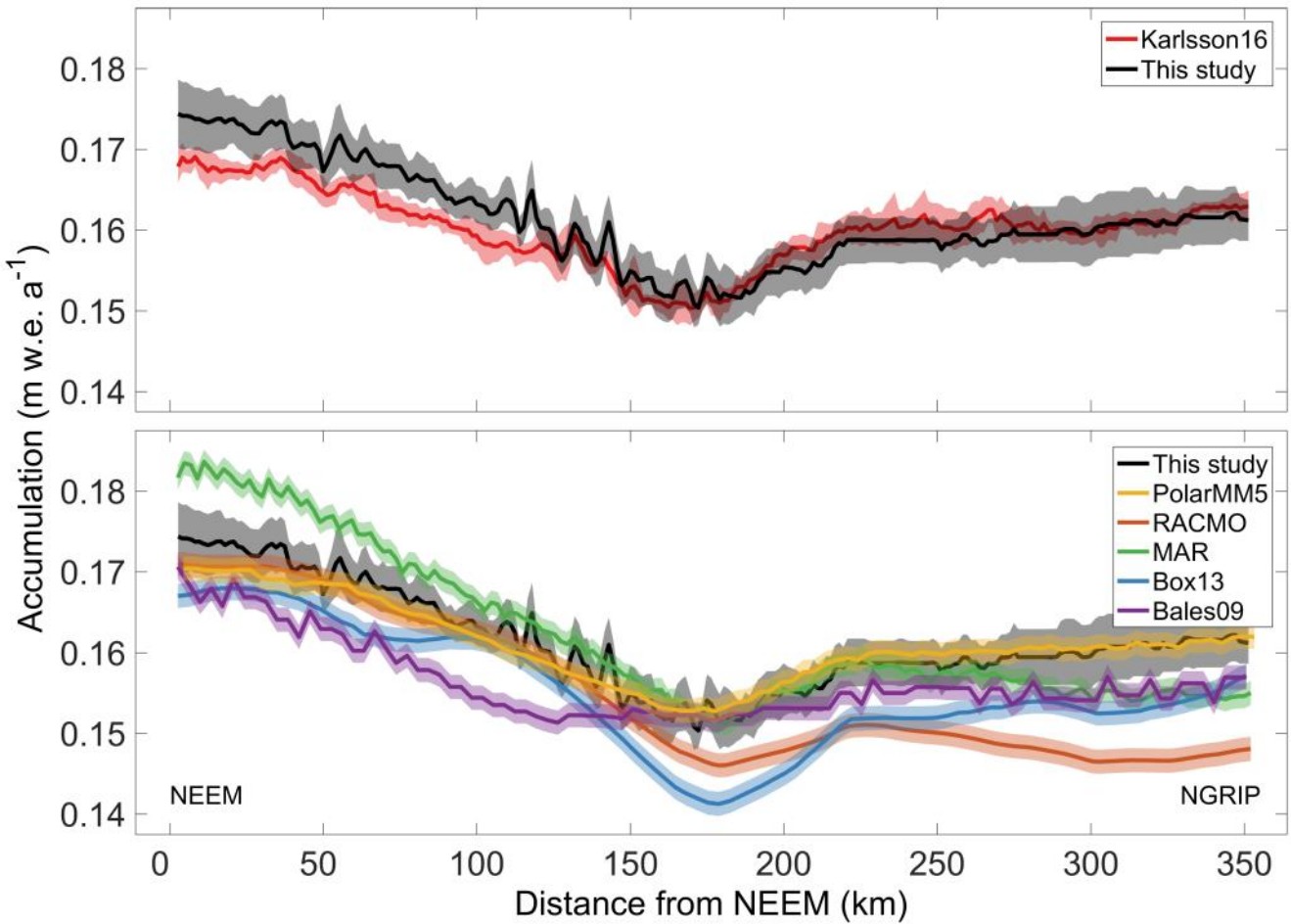

**Figure 9: (Top) Comparison of 1921-2014 IceBridge accumulation rates (this study) to 1911-2011 accumulation rates from Karlsson et al. (2016) along a transect from NEEM to NGRIP. On average, our measurements are 0.002 ± 0.002 m w.e. a$^{-1}$ higher than Karlsson16. (Bottom) Accumulation results (this study) compared with PolarMM5, RACMO, MAR, Box13, and Bales09 along the same transect.**

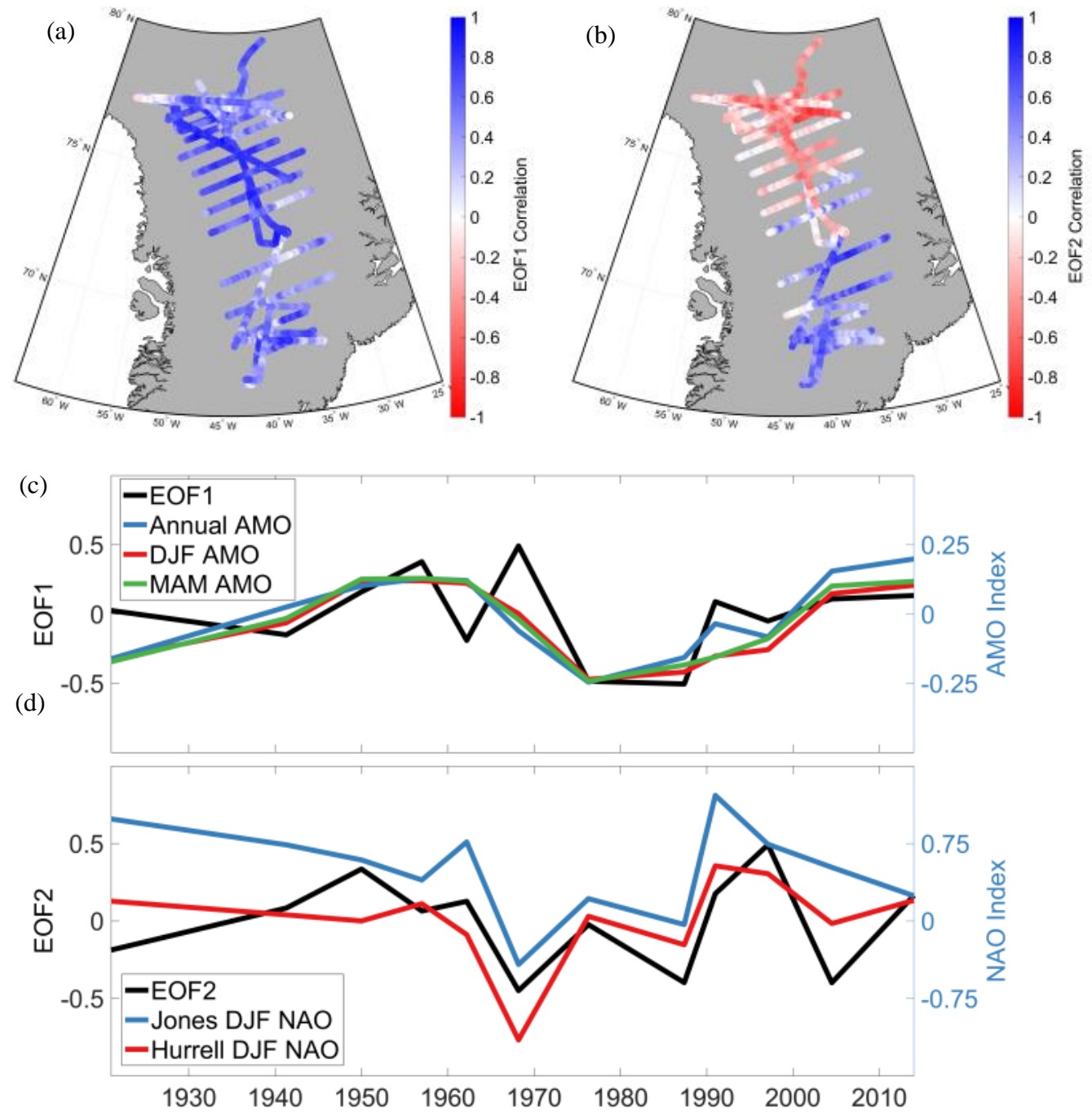

**Figure 10: Map of correlation between IceBridge accumulation and a) EOF1 and b) EOF2 of IceBridge accumulation data. c) EOF1 time series compared with the annually averaged, wintertime, and springtime Atlantic Meridional Oscillation (AMO) indices. d) EOF2 compared with the wintertime Hurrell (1995) and Jones (1997) North Atlantic Oscillation (NAO) indices.**

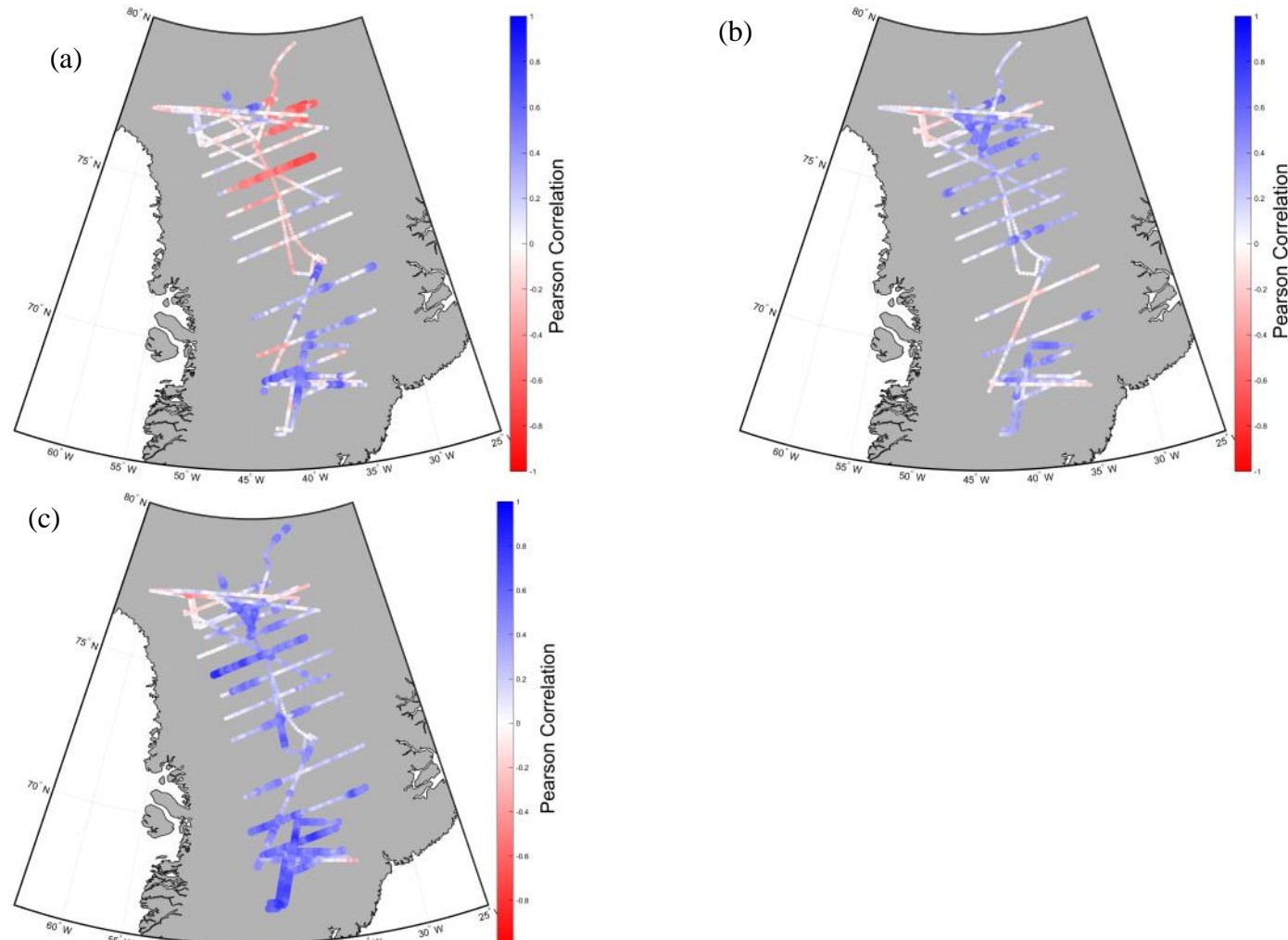

**Figure 11: Correlation map between 1899-2014 IceBridge accumulation and epoch-averaged climate indices. Statistically significant correlations (p < 0.05) are shown as larger data points. Maps show correlation of IceBridge data with a) Wintertime Jones (1997) NAO. b) Summer GBI. c) Annual AMO.**