# Peer review of "Regional Greenland Accumulation Variability from Operation IceBridge Airborne Accumulation Radar"

_The Cryosphere, 2016_

## Referee Comment (RC1) · Anonymous Referee #1 · 20 Nov 2016

This is a generally very good study of Greenland Ice Sheet accumulation based on Ice-Bridge data, that compares the results with several different regional climate models and a kriged map of ice-core data. Finally, an attempt is made to interpret recent accumulation variations (spatial and temporal) with reference to the Atlantic Multidecadal Oscillation and North Atlantic Oscillation changes, although Greenland Blocking should also be mentioned here. This latter section is less strong and can be supplemented with some extra material from recent studies (see below). I'm not convinced, from the results presented, that the AMO is necessarily the main driver of the Greenland accumulation increase seen since 1976, and would welcome a bit more analysis of this aspect. Overall the paper is important because it presents a major new dataset of

Greenland accumulation and highlights some major regional differences between the RCMs and IceBridge data, that need to be reconciled in future work. It helps to identify key regions where Greenland accumulation data are relatively lacking and need to be collected.

Specific comments Please use "GrIS" rather than "GIS" (Geographic Information Systems!) abbreviation for Greenland Ice Sheet. page 1, line 30: reference "Shepherd 2012" should be "Shepherd et al. 2012". I would add several further recent references here: Enderlin, E. M., I. M. Howat, S. Jeong,M.-J.Noh,J.H.vanAngelen,andM.R.van den Broeke (2014) An improvedmass budget for the Greenland icesheet, Geophys. Res. Lett., 41,866–872,doi:10.1002/2013GL059010. Hanna, E., F.J. Navarro, F. Pattyn, C. Domingues, X. Fettweis, E. Ivins, R.J. Nicholls, C. Ritz, B. Smith, S. Tulaczyk, P. Whitehouse & J. Zwally (2013) Ice-sheet mass balance and climate change. Nature 498, 51-59, doi: 10.1038/nature12238. van den Broeke, M. R., Enderlin, E. M., Howat, I. M., Kuipers Munneke, P., Noël, B. P. Y., van de Berg, W. J., van Meijgaard, E., and Wouters, B.: On the recent contribution of the Greenland ice sheet to sea level change, The Cryosphere, 10, 1933-1946, doi:10.5194/tc-10-1933-2016, 2016. p.2, l.3: supplement van den Broeke et al. (2009) reference with van den Broeke et al. (2016) (full details above). p.2, l.5 "due to complex relationships between accumulation variability and surface melt runoff" - add reference: Hanna, E., P. Huybrechts, I. Janssens, J. Cappelen, K. Steffen, and A. Stephens (2005), Runoff and mass balance of the Greenland ice sheet: 1958–2003, J. Geophys. Res., 110, D13108, doi:10.1029/2004JD005641. p.2, l.8: "preferred modes of climate variability like the NAO and AMO: add Greenland Blocking Index (GBI, Hanna et al. 2016) to these: Hanna, E., T. Cropper, R. Hall, J. Cappelen (2016) Greenland Blocking Index 1851-2015: a regional climate change signal. International Journal of Climatology, MS no. JOC-15-0742.R1, accepted/in press. p.2, l.13 Suggest add text in CAPS to the following: "but are too sparse to capture the full spatial variability of GIS accumulation, especially in the southeast," ALTHOUGH ATTEMPTS HAVE BEEN MADE TO INTERPOLATE ICE-CORE-BASED ACCUMULATION DATA - SUPPLEMENTED WITH COASTAL PRECIPITATION DATA - TO THE

WHOLE-ICE-SHEET SCALE (BALES ET AL. 2009). HOWEVER, THIS APPROACH MAY POSSIBLY UNDERESTIMATE ACCUMULATION IN PARTS OF THE INTERIOR COASTAL MOUNTAINS OF SOUTH-EAST GREENLAND. p.2, l.15 -> "more spatially distributed AND REPRESENTATIVE GIS accumulation dataset..." p.3, l.6 (and throughout MS) - correct "principle component analysis" to "principal component analysis". p.3, l.18: How are the IRHs related to spatial and/or temporal changes in accumulation? p.5, l.17, Equation 3: Is rho(z) the *mean* density of the respective layer? p.6, l.14: missing full stop . at end of sentence. p.8, l.21: "data set" -> "dataset". p.9, l.29: -> "where ice cores were collected several decades ago". p.9, l.31: "data poor regions" -> "data-poor regions". p.10, l.10: you can't really have a percentage of SMB as there is no absolute zero point, so I'm not sure this makes sense. p.10, l.26 slightly reword to "These correlations indicate AN ASSOCIATION BETWEEN the AMO AND Greenland precipitation ALTHOUGH, DUE TO COLLINEARITY, ANY PHYSICAL RELATION COULD PARTLY BE ACTING THROUGH NAO CHANGES." pp.10/11 overlap: Point out that the positive GrIS precipitation-AMO correlation, with warmer North Atlantic & Greenland temperatures, might also be due to associated storm-track or blocking changes (e.g. Hanna et al. 2013 IJOC, Hanna et al. 2016). Hanna, E., J.M. Jones, J. Cappelen, S.H. Mernild, L. Wood, K. Steffen & P. Huybrechts (2013) The influence of North Atlantic atmospheric and oceanic forcing effects on 1900–2010 Greenland summer climate and ice melt/runoff. Int. J. Climatol. 33, 862–880, doi: 10.1002/joc.3475. p.11, l.7 "Negative correlations in the northern and western regions...are indicative of greater precipitation during NAO negative conditions..." - but there should be positive correlations for Greenland overall (Greenland precip more generally reduces under negative NAO) because negative NAO is usually linked with positive GBI (anticyclonic conditions over Greenland, which should overall suppress precipitation) - please clarify. Obviously there are well-documented regional variations of this relation. p.11, l.25 -> "used to validate THE study". p.11, l.30 "we hypthesize that rising accumulation over most of the GIS interior since 1976 is related to an increasing AMO index" - rising accum. could equally well reflect changes in atmospheric circulation, e.g. a more

meridional airflow on average - with more moisture laden south-westerly winds, affecting Greenland. p.12, l.6: The Hanna et al. (2013) reference cited here should be for the IJOC paper referenced above, not the Nature paper - please amend. p.13, l.6 : change "strongest" to "most strongly". References Box & Rinke 2003 paper has the authors' names repeated twice. Please add other author names (or et al.) of the Shepherd 2012 Science paper. Table 1: add in the caption what the plus/minus figures represent.

---

## Referee Comment (RC2) · X. Fettweis (Referee) · 6 Dec 2016

This paper presents a new data set of the GrIS accumulation, which will be notably useful for validating RCM's in the dry snow zone. This paper fits well with TC, is well written and deserves to be published. However, before publication, a comparison with up-to-date RCM outputs will be more interesting and relevant, if it is not a too big job for the authors. The RCM outputs used here seem to be the ones used in Vernon et al. (2013)? Moreover, as already pointed by Reviewer #1, the discussion about the temporal variability is not enough scientifically robust to be published as it without additional validations.
* * *
1. This paper compares the Ice Bridge data set with outdated RCM outputs for them the accumulation biases highlighted in this paper (e.g. RACMO too dry and MAR too wet) were already identified and corrected in part (Noel et al., 2016; Fettweis et al., 2016). When MAR or RACMO outputs are shown, the used version of the RCMs should be at least mentioned. Are they the ones used in Vernon et al. (2013)? Last MARv3.5.2 outputs (used in Fettweis et al., 2016) can be found here: ftp://ftp.climato.be/fettweis/MARv3.5.2/Greenland/

Monthly outputs extrapolated at 5km can for example be found on ftp://ftp.climato.be/fettweis/MARv3.5.2/Greenland/NCEPv1_1948-2015_20km/monthly_outputs_interpolated_at_5km/

Idem for RACMO. The dry RACMO biases is now corrected in the 1km product based on the 11 km RACMO outputs (Noël et al, 2016). I am sure that Brice Noël will provide you these new outputs for an up-to-date comparison.

Finally, a comparison with Polar MM5 is shown although this model is not more used. A comparison with the Box et al. (2013) data set as done in (Fettweis et al., 2016) will be more interesting and relevant, if this does not request a too big job for the authors.

————————————————————————————————————————

2. My main critic is the discussion of the IceBridge temporal variability (in particular lines 21-34 of page 11) as already pointed out by Reviewer #1.

Firstly, I am very surprised that IceBridge does not see any trend in accumulation from 1712 to 1980's while for example, ice cores (Mernild et al., 2015), Box's reconstruction and MAR based outputs suggest a significant accumulation increase over 1920-1950's (as discussed in Fettweis et al., 2016) and at large scale, over the whole century (an increase of $0.1mm/yr^2$ over the last century is mentioned by Mernild et al. (2015)). How does the IceBridge data compare with the ice core based trends listed in Table 6 of Mernild et al. (2015)? The ice cores D2, D3, D4, D5,NEEM, NASA-U and SUMMIT

are in the IceBridge domain.

Secondly, how does the IceBridge interannual variability correlate with RCM outputs? Over 1976-2014? As discussed in Fettweis et al. (2016), the MAR interannual variability before 1950's can be questioning but over 1976-2014, the RCM interannual variability (fully driven by the renanalysis variability) is robust and should correlate with the IceBridge data set? How does the IceBridge interannual variability compare in respect to the RCM based one? The IceBridge signal seems to be very smoothed. Is there an interannual variability in the density used to extract the accumulation from the IceBridge signal?

Finally, the trends shown in Fig 11 are very small (only 1-2 mmWE/10yrs$^2$ !!) and are for me not enough significant to be mentioned according to the inter-annual variability ($\sim$30 mmWE/yrs) and the absolute value ($\sim$3000 mmWE/10 yrs). Such an accumulation increase over the recent decades has never been discussed/shown in previous studies and attributing these changes to AMO is dangerous (why does AMO not perturb accumulation over 1712-1980?). Below, there are the trend as well as the inter-annual variability simulated by MAR forced by NCEP1 over 1976-2014. Although MAR suggests rather a surface mass loss over the studied period, the MAR simulated trends are not significant in respect to the MAR based interannual variability (see attachment). Units are mmWE/yr$^2$ and not mmWE/10yr$^2$ !
* * *
SMB trend over 1976-2014 (mmWE/yr²)    SMB interannual variability over 1976-2014 (mmWE/yr)

**Fig. 1.**

---

## Short Comment (SC1) · 13 Dec 2016

This is a very nice study and it is good to see the accumulation radar datasets being put to use. We have just published a similar study also using Operation IceBridge data and a comparison between our two results would be very interesting. It could certainly give some insights into the robustness and reproducibility of this method for reconstructing past accumulation rates. For example, we retrieved higher than average accumulation rates during the past 100yr – does this dataset show the same? Our article can be found here: http://journal.frontiersin.org/article/10.3389/feart.2016.00097/full and the data here: http://www.iceandclimate.nbi.ku.dk/data/ or on the Pangaea website. Feel free to contact me if you need additional data for comparison.

Additionally, I would welcome some comments from the authors on a few points.

Age assignment

The IRHs are dated using an age-depth scale from a Summit core drilled in 2007. Presumably, when transferred to the radar data, this age-scale is corrected for the fact that 6-7 years of accumulation has been added in the time between the core was drilled and the radar data were acquired. However, this is not stated explicitly. Furthermore, it would be worth mentioning how far from the radar flightline this core was drilled. We found that even a few kilometres of distance between the radar data and our dated ice core (the NEEM 2011 S1 core) meant that the radar data had to be offset by 50ns due to snow accumulated around camp. At Summit, the snow has been moved around extensively due to camp activities and I would assume that a direct transfer from the Summit timescale to the radar data is not possible. Since our tests indicate that an error in dating of the order of +/- 15yr could lead to large changes in resulting accumulation rate (upwards of 20%) a more rigorous treatment/discussion of the dating of the IRHs is called.

Density profiles

The calibrated Herron-Langway model is only briefly described and the paragraph raises a few questions. E.g., where are the in-situ measurements of surface density located that were used to calibrate the model? How spatially variable are the densities in the data domain? The adjustment of +/- 5% to accumulation rates and surface density for testing the sensitivity of the final accumulation rates also seems a bit low. In our study, we retrieved the surface density using an inverse method, and found that the surface density varied by 2%. Our study area is in the dry snow zone at the ice divide, and therefore most likely less variable than the surface density closer to the margins. Furthermore, the difference between the IceBridge accumulation rates and the accumulation rates from Burgess et al. 2010 exceed 5% according to Table 2. Increasing the adjustment to 10% probably still returns accumulation rates well within the stated

uncertainty but it is worth a mention.

Layer thinning

It is not explicitly stated but I assume that the thinning of the layers due to downwards advection has been corrected for? If not then the accumulation rates are consistently underestimated for older layers. This would lead to a systematic bias in the result.

Ice flow

The manuscript contains no discussion on the influence of ice flow on the accumulation estimates. The movement of ice particles from, for example, low accumulation areas to high accumulation areas leads to an underestimation of the accumulation rate in the final measurement point. While 300yr is a relatively short time span and the ice particles have probably not moved very far there are some areas in the study region where ice flow velocities are high enough that it might have an impact. Again, this leads to a systematic over/underestimation rather than a random error.

Comparison with ice core / shallow core measurements

The IceBridge accumulation rates are compared to ice core accumulation "over the time domain of each ice core". Does this imply that the IceBridge accumulation rates are compared to accumulation rates from the entire NEEM core including the last glacial period? I would assume not but a direct indication of which time periods are used for this comparison would be very helpful. I also wonder why cores from the central and northern part of Greenland are not included? For example, accumulation rates from the NGRIP core and from the cores B26 and B29. Accumulation rates from the latter two were recently published by Weissbach et al., 2016, Climate of the Past.

I realise that some of these effects might lead to smaller uncertainties than the stated uncertainty in accumulation rate of +/- 0.127 m w.e./yr. However, by not mentioning these complications a reader could get the impression that no such complications exist.

Finally, a comparison between the uncertainty with the accumulation rates from Fig.

5 indicates that this corresponds to at least 25% (highest accumulation rate is 0.5 m w.e./yr) or more for lower accumulation rate areas. In that context, how can the % differences in Table 2 be significant even at less than 10% difference? Am I missing something here?

Figures

Figure 2: The label on the colorbar says "Age of oldest layer" but presumably, it should say date of the IRH (in Common Era)? Figure 3: This figure implies that the accumulation rate was calculated as an average for the period 1712-2014 but from Figure 2 parts of these lines do not have layers that go that far back in time.

I apologise if this echoes any comments from reviewer #2. I wrote this before the second review had been posted.

---

## Author Comment (AC1) · 7 Feb 2017

Comment: This is a very nice study and it is good to see the accumulation radar datasets being put to use. We have just published a similar study also using Operation IceBridge data and a comparison between our two results would be very interesting. It could certainly give some insights into the robustness and reproducibility of this method for reconstructing past accumulation rates. For example, we retrieved higher than average accumulation rates during the past 100yr – does this dataset show the same? Our article can be found here: http://journal.frontiersin.org/article/10.3389/feart.2016.00097/full and the data here: http://www.iceandclimate.nbi.ku.dk/data/ or on the Pangaea website. Feel free to contact me if you need additional data for comparison.

Response: We have added a section in the manuscript comparing our results to those in Karlsson et al. (2016). We find very similar values across the transect from NEEM to NGRIP, however we find slightly higher values during the 19th century (instead of the 20th century) than the 1712-2014 average. These differences are very small (<0.01 m w.e. a-1) and are most likely due to differences in our estimated depth-density profiles. We have added a figure comparing our accumulation measurements to the Karlsson et al. (2016) measurements.

"Comparison with Karlsson et al., 2016 A study by Karlsson et al. (2016; hereafter Karlsson16) uses a very different method to calculate accumulation from IceBridge Accumulation Radar data near NEEM and NGRIP. We compare data from their study, representing flight lines in 2011 and 2012, to a repeat flight during the 2014 IceBridge season analyzed using our method. In Figure 10 [reproduced below], the 1921-2014 accumulation rates (this study) are plotted against 1911-2011 Karlsson16 accumulation rates and the RCMs used for comparison in this study. On average along the 350 km flight line, the accumulation rates calculated in this study are 0.002 ± 0.005 m w.e. a-1 higher than in Karlsson16, well within calculated error, and in better agreement than the RCMs. Our accumulation values agree better with Karlsson16 from 150 km along the transect to NGRIP (underestimate of 0.002 ± 0.002 m w.e. a-1) than they do along the first half of the transect (overestimate of 0.007 ± 0.004 m w.e a-1). The average 1817-1921 measurements are 0.01 m w.e. a-1 higher than the 1811-1911 Karlsson16 values, and the 1712-1811 measurements are 0.0081 m w.e. a-1 higher than the 1711-1811 Karlsson16 values. Thus, our results are nearly identical with Karlsson16 over the time domain of this study, despite the two studies using different methods to calculate accumulation, analyzing different IceBridge flights from different years, and tracing IRHs from different ice cores."

Comment: Additionally, I would welcome some comments from the authors on a few points. Age assignment The IRHs are dated using an age-depth scale from a Summit

core drilled in 2007. Presumably, when transferred to the radar data, this age-scale is corrected for the fact that 6-7 years of accumulation has been added in the time between the core was drilled and the radar data were acquired. However, this is not stated explicitly. Furthermore, it would be worth mentioning how far from the radar flightline this core was drilled. We found that even a few kilometres of distance between the radar data and our dated ice core (the NEEM 2011 S1 core) meant that the radar data had to be offset by 50ns due to snow accumulated around camp. At Summit, the snow has been moved around extensively due to camp activities and I would assume that a direct transfer from the Summit timescale to the radar data is not possible. Since our tests indicate that an error in dating of the order of +/- 15yr could lead to large changes in resulting accumulation rate (upwards of 20%) a more rigorous treatment/discussion of the dating of the IRHs is called.

Response: We added the following sentences: "These ice cores are 3 and 7 km from the closest IceBridge radar trace, so we assume similar accumulation rates across this small distance. We correct for the 7-year difference between ice core collection and IceBridge radar flights by extrapolating the depth-age curve."

Comment: Density profiles The calibrated Herron-Langway model is only briefly described and the paragraph raises a few questions. E.g., where are the in-situ measurements of surface density located that were used to calibrate the model? How spatially variable are the densities in the data domain? The adjustment of +/- 5% to accumulation rates and surface density for testing the sensitivity of the final accumulation rates also seems a bit low. In our study, we retrieved the surface density using an inverse method, and found that the surface density varied by 2%. Our study area is in the dry snow zone at the ice divide, and therefore most likely less variable than the surface density closer to the margins. Furthermore, the difference between the IceBridge accumulation rates and the accumulation rates from Burgess et al. 2010 exceed 5% according to Table 2. Increasing the adjustment to 10% probably still returns accumulation rates well within the stated uncertainty but it is worth a mention.

Response: Snow density values are obtained from ice cores at Summit, field measurements at EGIG t-31, and shallow cores at PARCA locations to calibrate Herron-Langway model. Elsewhere we use snow density values from MAR. We have now clarified these data sources in the text.

Comment: Layer thinning It is not explicitly stated but I assume that the thinning of the layers due to downwards advection has been corrected for? If not then the accumulation rates are consistently underestimated for older layers. This would lead to a systematic bias in the result.

Response: We have corrected for layer thinning and added the following paragraph: "The effect of layer thinning is very small for the time domain and region of this study. However, we correct for layer thinning due to downward advection using a Nye (1963) model. For each radar trace, the thinning factor, ïĄň(z), is calculated from the average accumulation, b ÌĞ (m w.e. a-1), age of the IRH, a (year), and thickness of the GrIS, H (m), from Morlighem et al. (2014): $\lambda(z)=e^{(-b ÌĞ/H a)}$."

Comment: Ice flow The manuscript contains no discussion on the influence of ice flow on the accumulation estimates. The movement of ice particles from, for example, low accumulation areas to high accumulation areas leads to an underestimation of the accumulation rate in the final measurement point. While 300yr is a relatively short time span and the ice particles have probably not moved very far there are some areas in the study region where ice flow velocities are high enough that it might have an impact. Again, this leads to a systematic over/underestimation rather than a random error.

Response: We added the following paragraph: "We do not correct for ice flow due to advection of the ice sheet since nearly all of the radar traces occur in areas with surface velocities < 50 m a-1. The only areas with higher velocities are across NEGIS and one small area in the southwest. Velocities in these areas are $\sim$ 60-100 m a-1 for the time domain of this study and do not significantly affect accumulation results"

Comment: Comparison with ice core / shallow core measurements The IceBridge accumulation rates are compared to ice core accumulation "over the time domain of each ice core". Does this imply that the IceBridge accumulation rates are compared to accumulation rates from the entire NEEM core including the last glacial period? I would assume not but a direct indication of which time periods are used for this comparison would be very helpful. I also wonder why cores from the central and northern part of Greenland are not included? For example, accumulation rates from the NGRIP core and from the cores B26 and B29. Accumulation rates from the latter two were recently published by Weissbach et al., 2016, Climate of the Past.

Response: The IceBridge accumulation rates are compared to ice core accumulation over corresponding time domains; we have added a column in Table 1 to more clearly reflect the time ranges. Accumulation rates and trends are also now calculated for B26, B29, and NGRIP and presented in Table 2.

Comment: I realise that some of these effects might lead to smaller uncertainties than the stated uncertainty in accumulation rate of +/- 0.127 m w.e./yr. However, by not mentioning these complications a reader could get the impression that no such complications exist.

Finally, a comparison between the uncertainty with the accumulation rates from Fig. 5 indicates that this corresponds to at least 25% (highest accumulation rate is 0.5 m w.e./yr) or more for lower accumulation rate areas. In that context, how can the % differences in Table 2 be significant even at less than 10% difference? Am I missing something here?

Response: The IceBridge accumulation rates in Figure 5 are statistically indistinguishable from ice core accumulation rates, for each core. Moreover, the uncertainty decreases when IceBridge accumulation rates are averaged over 1957-2014, as used for comparison in Table 2. This uncertainty decreases since small errors in layer tracing become less important when calculating accumulation over longer timespans.

Comment: Figures Figure 2: The label on the colorbar says "Age of oldest layer" but

presumably, it should say date of the IRH (in Common Era)?

Response: We changed the caption to "Date of oldest resolvable IRH" for clarity.

Comment: Figure 3: This figure implies that the accumulation rate was calculated as an average for the period 1712-2014 but from Figure 2 parts of these lines do not have layers that go that far back in time.

Response: The accumulation rates for Figure 3 are averaged over the entire time domain for each radar trace. The caption has been corrected to "Average accumulation over the temporal domain of each radar trace calculated from IceBridge…"

Comment: I apologise if this echoes any comments from reviewer #2. I wrote this before the second review had been posted.

Response: Thank you for your comments.

Figure 10: (Top) Comparison of 1921-2014 IceBridge accumulation rates (this study) to 1911-2011 accumulation rates from Karlsson et al. (2016) along a transect from NEEM to NGRIP. On average, our measurements are $0.002 \pm 0.002$ m w.e. a-1 higher than Karlsson16. (Bottom) Accumulation results (this study) compared with PolarMM5, RACMO, MAR, Box13, and Bales09 along the same transect.
* * *
[Figure]

[Figure]

[Figure]

**Fig. 1.**

---

## Author Comment (AC2) · 7 Feb 2017

Comment: This paper presents a new data set of the GrIS accumulation, which will be notably useful for validating RCM's in the dry snow zone. This paper fits well with TC, is well written and deserves to be published. However, before publication, a comparison with up-to-date RCM outputs will be more interesting and relevant, if it is not a too big job for the authors. The RCM outputs used here seem to be the ones used in Vernon et al. (2013)? Moreover, as already pointed by Reviewer #1, the discussion about the temporal variability is not enough scientifically robust to be published as it without additional validations.

Response: The most recent RCM outputs have now been used for this study. We now

compare IceBridge accumulation with MAR v3.5.2 and RACMO2.3.

Comment: 1. This paper compares the Ice Bridge data set with outdated RCM outputs for them the accumulation biases highlighted in this paper (e.g. RACMO too dry and MAR too wet) were already identified and corrected in part (Noel et al., 2016; Fettweis et al., 2016). When MAR or RACMO outputs are shown, the used version of the RCMs should be at least mentioned. Are they the ones used in Vernon et al. (2013)? Last MARv3.5.2 outputs (used in Fettweis et al., 2016) can be found here: ftp://ftp.climato.be/fettweis/MARv3.5.2/Greenland/ Monthly outputs extrapolated at 5km can for example be found on ftp://ftp.climato.be/fettweis/MARv3.5.2/Greenland/NCEPv1_1948-2015_20km/monthly_outputs_interpolated_at_5km/ Idem for RACMO. The dry RACMO biases is now corrected in the 1km product based on the 11 km RACMO outputs (Noël et al, 2016). I am sure that Brice Noël will provide you these new outputs for an up-to-date comparison.

Response: We now use the most recent model outputs.

Comment: Finally, a comparison with Polar MM5 is shown although this model is not more used. A comparison with the Box et al. (2013) data set as done in (Fettweis et al., 2016) will be more interesting and relevant, if this does not request a too big job for the authors.

Response: We now include accumulation data from Box et al. (2013).

Comment: 2. My main critic is the discussion of the IceBridge temporal variability (in particular lines 21-34 of page 11) as already pointed out by Reviewer #1.

Response: We have addressed the reviewers' concerns about the temporal variability discussion. Please see response to Reviewer #1.

Comment: Firstly, I am very surprised that IceBridge does not see any trend in accumulation from 1712 to 1980's while for example, ice cores (Mernild et al., 2015), Box's

reconstruction and MAR based outputs suggest a significant accumulation increase over 1920-1950's (as discussed in Fettweis et al., 2016) and at large scale, over the whole century (an increase of 0.1mm/yr2 over the last century is mentioned by Mernild et al. (2015)).

Response: IceBridge does see trends in accumulation from 1712-1980, but they are not as large nor as significant as the later trends. We have added IceBridge accumulation trends to Table 1.

Comment: How does the IceBridge data compare with the ice core based trends listed in Table 6 of Mernild et al. (2015)? The ice cores D2, D3, D4, D5,NEEM, NASA-U and SUMMIT are in the IceBridge domain.

Response: We calculated accumulation trends for ice cores and nearest IceBridge radar traces in Table 1 of this study. The IceBridge and ice core trends agree within error for the corresponding time periods. Likewise, accumulation trends agree with those from Mernild et al. (2015) for longer time periods, as reflected in Table 1. With our 15-30 year temporal resolution over the 20th century, it is difficult to directly compare with 30-year trends from Mernild et al. (2015).

Comment: Secondly, how does the IceBridge interannual variability correlate with RCM outputs? Over 1976-2014? As discussed in Fettweis et al. (2016), the MAR interannual variability before 1950's can be questioning but over 1976-2014, the RCM interannual variability (fully driven by the renanalysis variability) is robust and should correlate with the IceBridge data set? How does the IceBridge interannual variability compare in respect to the RCM based one? The IceBridge signal seems to be very smoothed. Is there an interannual variability in the density used to extract the accumulation from the IceBridge signal?

Response: Similar to the RCMs, we see less accumulation variability towards the interior of the ice sheet than towards the south and the coasts. Our calculated variability is the same order of magnitude as RCM variability (0.03-0.06 m w.e a-1 in the northern interior and 0.1-0.15 m w.e a-1 in the south), however our variability is not nearly as smooth since we are calculating variability using internal reflection horizons form airborne radar. We did not perform an extensive statistical review on these data since they agree with RCMs and our expectations. See figure below. There is no interannual variability in the density used to extract accumulation. We use a steady state Herron-Langway (1980) profile to drive our depth-density model.

Comment: Finally, the trends shown in Fig. 11 are very small (only 1-2 mmWE/10yrs2 !!) and are for me not enough significant to be mentioned according to the inter-annual variability (∼30 mmWE/yrs) and the absolute value (∼3000 mmWE/10 yrs). Such an accumulation increase over the recent decades has never been discussed/shown in previous studies and attributing these changes to AMO is dangerous (why does AMO not perturb accumulation over 1712-1980?). Below, there are the trend as well as the interannual variability simulated by MAR forced by NCEP1 over 1976-2014. Although MAR suggests rather a surface mass loss over the studied period, the MAR simulated trends are not significant in respect to the MAR based interannual variability (see attachment). Units are mmWE/yr2 and not mmWE/10yr2 !

Response: The reviewer's point is well taken. The trends shown in original Fig. 11 were not statistically significant, so we have removed this figure and related discussion from the manuscript.

Figure: SMB interannual variability calculated as one standard deviation of 1921-2014 accumulation for each radar trace.

**Fig. 1.**

Accumulation Variability (m w.e. a$^{-1}$)

---

## Author Comment (AC3) · 7 Feb 2017

Comment: This is a generally very good study of Greenland Ice Sheet accumulation based on Ice-Bridge data, that compares the results with several different regional climate models and a kriged map of ice-core data. Finally, an attempt is made to interpret recent accumulation variations (spatial and temporal) with reference to the Atlantic Multidecadal Oscillation and North Atlantic Oscillation changes, although Greenland Blocking should also be mentioned here. This latter section is less strong and can be supplemented with some extra material from recent studies (see below). I'm not convinced, from the results presented, that the AMO is necessarily the main driver of the Greenland accumulation increase seen since 1976, and would welcome a bit more

analysis of this aspect. Overall the paper is important because it presents a major new dataset of Greenland accumulation and highlights some major regional differences between the RCMs and IceBridge data, that need to be reconciled in future work. It helps to identify key regions where Greenland accumulation data are relatively lacking and need to be collected.

Response: We have significantly modified the portion of the manuscript evaluating spatial and temporal variations in accumulation and their relationships with atmospheric and oceanic modes of variability. We expand the discussion of relationships with the NAO, AMO and GBI, and incorporate additional relevant references. These results are consistent with our original EOF analysis, but we think that our new discussion and figure based on correlations significantly improves the manuscript.

Comment: Specific comments: Please use "GrIS" rather than "GIS" (Geographic Information Systems!) abbreviation for Greenland Ice Sheet.

Response: This acronym has been corrected to GrIS everywhere in the paper.

Comment: page 1, line 30: reference "Shepherd 2012" should be "Shepherd et al. 2012". I would add several further recent references here: Enderlin, E. M., I. M. Howat, S. Jeong,M.-J.Noh,J.H.vanAngelen,and M.R.van den Broeke (2014) An improved mass budget for the Greenland icesheet, Geophys. Res. Lett., 41,866–872,doi:10.1002/2013GL059010. Hanna, E., F.J. Navarro, F. Pattyn, C. Domingues, X. Fettweis, E. Ivins, R.J. Nicholls, C. Ritz, B. Smith, S. Tulaczyk, P. Whitehouse & J. Zwally (2013) Ice-sheet mass balance and climate change. Nature 498, 51-59, doi: 10.1038/nature12238. van den Broeke, M. R., Enderlin, E. M., Howat, I. M., Kuipers Munneke, P., Noël, B. P. Y., van de Berg, W. J., van Meijgaard, E., and Wouters, B.: On the recent contribution of the Greenland ice sheet to sea level change, The Cryosphere, 10, 1933-1946, doi:10.5194/tc-10-1933-2016, 2016.

Response: The reference has been corrected and additional references have been added.

Comment: p.2, l.3: supplement van den Broeke et al. (2009) reference with van den Broeke et al. (2016) (full details above).

Response: The reference has been added.

Comment: p.2, l.5 "due to complex relationships between accumulation variability and surface melt runoff" - add reference: Hanna, E., P. Huybrechts, I. Janssens, J. Cappelen, K. Steffen, and A. Stephens (2005), Runoff and mass balance of the Greenland ice sheet: 1958–2003, J. Geophys. Res., 110, D13108, doi:10.1029/2004JD005641.

Response: The reference has been added.

Comment: p.2, l.8: "preferred modes of climate variability like the NAO and AMO: add Greenland Blocking Index (GBI, Hanna et al. 2016) to these: Hanna, E., T. Cropper, R. Hall, J. Cappelen (2016) Greenland Blocking Index 1851-2015: a regional climate change signal. International Journal of Climatology, MS no. JOC-15-0742.R1, accepted/in press.

Response: This text and reference have been added.

Comment: p.2, l.13 Suggest add text in CAPS to the following: "but are too sparse to capture the full spatial variability of GIS accumulation, especially in the southeast," ALTHOUGH ATTEMPTS HAVE BEEN MADE TO INTERPOLATE ICE-CORE-BASED ACCUMULATION DATA - SUPPLEMENTED WITH COASTAL PRECIPITATION DATA - TO THE WHOLE-ICE-SHEET SCALE (BALES ET AL. 2009). HOWEVER, THIS APPROACH MAY POSSIBLY UNDERESTIMATE ACCUMULATION IN PARTS OF THE INTERIOR COASTAL MOUNTAINS OF SOUTH-EAST GREENLAND.

Response: The suggested changes have been made.

Comment: p.2, l.15 -> "more spatially distributed AND REPRESENTATIVE GIS accumulation dataset..."

Response: The suggested changes have been made.

[Figure]

Comment: p.3, l.6 (and throughout MS) - correct "principle component analysis" to "principal component analysis".

Response: The suggested changes have been made.

Comment: p.3, l.18: How are the IRHs related to spatial and/or temporal changes in accumulation?

Response: We have added the following text: "We calculate accumulation between each pair of adjacent IRHs for every radar trace along the flight lines. Spatial changes in accumulation are evident from varying distances between IRHs along each flight line. Temporal changes in accumulation are evident from examining accumulation during different epochs at one location."

Comment: p.5, l.17, Equation 3: Is rho(z) the *mean* density of the respective layer?

Response: Yes, rho(z) is the mean density between IRHs. This has been clarified.

Comment: p.6, l.14: missing full stop at end of sentence.

Response: The suggested changes have been made.

Comment: p.8, l.21: "data set" -> "dataset".

Response: The suggested changes have been made.

Comment: p.9, l.29: ->"where ice cores were collected several decades ago".

Response: The suggested changes have been made.

Comment: p.9, l.31: "data poor regions" -> "data-poor regions".

Response: The suggested changes have been made.

Comment: p.10, l.10: you can't really have a percentage of SMB as there is no absolute zero point, so I'm not sure this makes sense.

Response: These are accumulation percent differences calculated using (Model – Ice-

Bridge)/IceBridge, which we use extensively in Figure 8 and Table 2.

Comment: p.10, l.26 slightly reword to "These correlations indicate AN ASSOCIATION BETWEEN the AMO AND Greenland precipitation ALTHOUGH, DUE TO COLLINEAR-ITY, ANY PHYSICAL RELATION COULD PARTLY BE ACTING THROUGH NAO CHANGES."

Response: The suggested changes have been made.

Comment: pp.10/11 overlap: Point out that the positive GrIS precipitation-AMO correlation, with warmer North Atlantic & Greenland temperatures, might also be due to associated storm-track or blocking changes (e.g. Hanna et al. 2013 IJOC, Hanna et al. 2016). Hanna, E., J.M. Jones, J. Cappelen, S.H. Mernild, L. Wood, K. Steffen & P. Huybrechts (2013) The influence of North Atlantic atmospheric and oceanic forcing effects on 1900–2010 Greenland summer climate and ice melt/runoff. Int. J. Climatol. 33, 862–880, doi: 10.1002/joc.3475.

Response: The suggested references to storm-tracks and the GBI have now been included in our revised section on accumulation relationships with climate modes.

Comment: p.11, l.7 "Negative correlations in the northern and western regions...are indicative of greater precipitation during NAO negative conditions..." - but there should be positive correlations for Greenland overall (Greenland precip more generally reduces under negative NAO) because negative NAO is usually linked with positive GBI (anticyclonic conditions over Greenland, which should overall suppress precipitation) - please clarify. Obviously there are well-documented regional variations of this relation.

Response: The confusion here may stem from the seasonality of the NAO correlations (winter) and the GBI correlations (primarily summer, when the GBI and NAO indices show the largest differences). Our new figure [reproduced below] using seasonal-annual correlations with the NAO, GBI and AMO clarifies these relationships. Our analysis of EOF2 in the context of the wintertime NAO is supported by the similar

temporal variability of the wintertime NAO index and EOF time series, and the strong similarity of the EOF2 vs. IceBridge accumulation correlation map to the wintertime NAO vs. IceBridge accumulation correlation map. Our new figure shows a weak positive correlation between IceBridge accumulation and summertime GBI – i.e. slightly higher accumulation during enhanced blocking. While this may seem counter-intuitive, this relationship is driven by enhanced meridional flow and moisture advection into Greenland under the weak zonal flow associated with GBI positive (NAO negative) conditions (Hanna et al., 2016). We respectfully disagree that there should be positive correlations between Greenland precipitation and the NAO overall. Box et al. (2013) found that the sign of this correlation reverses four times from 1880-2005. Hanna et al. (2011) found no significant correlation between the NAO and Greenland-wide precipitation from 1870-2009 and 1950-2009 (their Table 7).

Comment: p.11, l.25 -> "used to validate THE study".

Response: The suggested changes have been made.

Comment: p.11, l.30 "we hypthesize that rising accumulation over most of the GIS interior since 1976 is related to an increasing AMO index" – rising accum. could equally well reflect changes in atmospheric circulation, e.g. a more meridional airflow on average - with more moisture laden south-westerly winds, affecting Greenland.

Response: We no longer discuss this recent accumulation rise because it is not statistically significant. The relevant figure (original Figure 11) has also been removed.

Comment: p.12, l.6: The Hanna et al. (2013) reference cited here should be for the IJOC paper referenced above, not the Nature paper - please amend.

Response: The suggested changes have been made.

Comment: p.13, l.6 : change "strongest" to "most strongly". References Box & Rinke 2003 paper has the authors' names repeated twice. Please add other author names (or et al.) of the Shepherd 2012 Science paper.

Response: The suggested changes have been made.

Comment: Table 1: add in the caption what the plus/minus figures represent.

Response: The suggested changes have been made.

Figure 12: Correlation map between 1899-2014 IceBridge accumulation and epoch-averaged climate indices. Statistically significant correlations are shown as larger data points. Maps show correlation of IceBridge data with a) Wintertime Jones (1997) NAO. b) Annual Jones (1997) NAO. c) Summer GBI. d) Annual AMO.
* * *
[Figure]

**Fig. 1.**

---

## Author Comment (AC4) · 7 Feb 2017

Comment: This is a generally very good study of Greenland Ice Sheet accumulation based on Ice-Bridge data, that compares the results with several different regional climate models and a kriged map of ice-core data. Finally, an attempt is made to interpret recent accumulation variations (spatial and temporal) with reference to the Atlantic Multidecadal Oscillation and North Atlantic Oscillation changes, although Greenland Blocking should also be mentioned here. This latter section is less strong and can be supplemented with some extra material from recent studies (see below). I'm not convinced, from the results presented, that the AMO is necessarily the main driver of the Greenland accumulation increase seen since 1976, and would welcome a bit more

analysis of this aspect. Overall the paper is important because it presents a major new dataset of Greenland accumulation and highlights some major regional differences between the RCMs and IceBridge data, that need to be reconciled in future work. It helps to identify key regions where Greenland accumulation data are relatively lacking and need to be collected.

Response: We have significantly modified the portion of the manuscript evaluating spatial and temporal variations in accumulation and their relationships with atmospheric and oceanic modes of variability. We expand the discussion of relationships with the NAO, AMO and GBI, and incorporate additional relevant references. These results are consistent with our original EOF analysis, but we think that our new discussion and figure based on correlations significantly improves the manuscript.

Comment: Specific comments: Please use "GrIS" rather than "GIS" (Geographic Information Systems!) abbreviation for Greenland Ice Sheet.

Response: This acronym has been corrected to GrIS everywhere in the paper.

Comment: page 1, line 30: reference "Shepherd 2012" should be "Shepherd et al. 2012". I would add several further recent references here: Enderlin, E. M., I. M. Howat, S. Jeong,M.-J.Noh,J.H.vanAngelen,and M.R.van den Broeke (2014) An improved mass budget for the Greenland icesheet, Geophys. Res. Lett., 41,866–872,doi:10.1002/2013GL059010. Hanna, E., F.J. Navarro, F. Pattyn, C. Domingues, X. Fettweis, E. Ivins, R.J. Nicholls, C. Ritz, B. Smith, S. Tulaczyk, P. Whitehouse & J. Zwally (2013) Ice-sheet mass balance and climate change. Nature 498, 51-59, doi: 10.1038/nature12238. van den Broeke, M. R., Enderlin, E. M., Howat, I. M., Kuipers Munneke, P., Noël, B. P. Y., van de Berg, W. J., van Meijgaard, E., and Wouters, B.: On the recent contribution of the Greenland ice sheet to sea level change, The Cryosphere, 10, 1933-1946, doi:10.5194/tc-10-1933-2016, 2016.

Response: The reference has been corrected and additional references have been added.

Comment: p.2, l.3: supplement van den Broeke et al. (2009) reference with van den Broeke et al. (2016) (full details above).

Response: The reference has been added.

Comment: p.2, l.5 "due to complex relationships between accumulation variability and surface melt runoff" - add reference: Hanna, E., P. Huybrechts, I. Janssens, J. Cappelen, K. Steffen, and A. Stephens (2005), Runoff and mass balance of the Greenland ice sheet: 1958–2003, J. Geophys. Res., 110, D13108, doi:10.1029/2004JD005641.

Response: The reference has been added.

Comment: p.2, l.8: "preferred modes of climate variability like the NAO and AMO: add Greenland Blocking Index (GBI, Hanna et al. 2016) to these: Hanna, E., T. Cropper, R. Hall, J. Cappelen (2016) Greenland Blocking Index 1851-2015: a regional climate change signal. International Journal of Climatology, MS no. JOC-15-0742.R1, accepted/in press.

Response: This text and reference have been added.

Comment: p.2, l.13 Suggest add text in CAPS to the following: "but are too sparse to capture the full spatial variability of GIS accumulation, especially in the southeast," ALTHOUGH ATTEMPTS HAVE BEEN MADE TO INTERPOLATE ICE-CORE-BASED ACCUMULATION DATA - SUPPLEMENTED WITH COASTAL PRECIPITATION DATA - TO THE WHOLE-ICE-SHEET SCALE (BALES ET AL. 2009). HOWEVER, THIS APPROACH MAY POSSIBLY UNDERESTIMATE ACCUMULATION IN PARTS OF THE INTERIOR COASTAL MOUNTAINS OF SOUTH-EAST GREENLAND.

Response: The suggested changes have been made.

Comment: p.2, l.15 -> "more spatially distributed AND REPRESENTATIVE GIS accumulation dataset..."

Response: The suggested changes have been made.

Comment: p.3, l.6 (and throughout MS) - correct "principle component analysis" to "principal component analysis".

Response: The suggested changes have been made.

Comment: p.3, l.18: How are the IRHs related to spatial and/or temporal changes in accumulation?

Response: We have added the following text: "We calculate accumulation between each pair of adjacent IRHs for every radar trace along the flight lines. Spatial changes in accumulation are evident from varying distances between IRHs along each flight line. Temporal changes in accumulation are evident from examining accumulation during different epochs at one location."

Comment: p.5, l.17, Equation 3: Is rho(z) the *mean* density of the respective layer?

Response: Yes, rho(z) is the mean density between IRHs. This has been clarified.

Comment: p.6, l.14: missing full stop at end of sentence.

Response: The suggested changes have been made.

Comment: p.8, l.21: "data set" -> "dataset".

Response: The suggested changes have been made.

Comment: p.9, l.29: ->"where ice cores were collected several decades ago".

Response: The suggested changes have been made.

Comment: p.9, l.31: "data poor regions" -> "data-poor regions".

Response: The suggested changes have been made.

Comment: p.10, l.10: you can't really have a percentage of SMB as there is no absolute zero point, so I'm not sure this makes sense.

Response: These are accumulation percent differences calculated using (Model – Ice-

Bridge)/IceBridge, which we use extensively in Figure 8 and Table 2.

Comment: p.10, l.26 slightly reword to "These correlations indicate AN ASSOCIATION BETWEEN the AMO AND Greenland precipitation ALTHOUGH, DUE TO COLLINEAR-ITY, ANY PHYSICAL RELATION COULD PARTLY BE ACTING THROUGH NAO CHANGES."

Response: The suggested changes have been made.

Comment: pp.10/11 overlap: Point out that the positive GrIS precipitation-AMO cor-relation, with warmer North Atlantic & Greenland temperatures, might also be due to associated storm-track or blocking changes (e.g. Hanna et al. 2013 IJOC, Hanna et al. 2016). Hanna, E., J.M. Jones, J. Cappelen, S.H. Mernild, L. Wood, K. Steffen & P. Huybrechts (2013) The influence of North Atlantic atmospheric and oceanic forcing effects on 1900–2010 Greenland summer climate and ice melt/runoff. Int. J. Climatol. 33, 862–880, doi: 10.1002/joc.3475.

Response: The suggested references to storm-tracks and the GBI have now been included in our revised section on accumulation relationships with climate modes.

Comment: p.11, l.7 "Negative correlations in the northern and western regions...are in-dicative of greater precipitation during NAO negative conditions..." - but there should be positive correlations for Greenland overall (Greenland precip more generally reduces under negative NAO) because negative NAO is usually linked with positive GBI (anticy-clonic conditions over Greenland, which should overall suppress precipitation) - please clarify. Obviously there are well-documented regional variations of this relation.

Response: We respectfully disagree that there should be positive correlations between Greenland precipitation and the NAO overall. Box et al. (2013) found that the sign of this correlation reverses four times from 1880-2005. Hanna et al. (2011) found no significant correlation between the NAO and Greenland-wide precipitation from 1870-2009 and 1950-2009 (their Table 7). Our new figure using seasonal-annual correlations

with the NAO, GBI and AMO clarifies their relationships with IB accumulation.

Our analysis of EOF2 in the context of the wintertime NAO is supported by the similar temporal variability of the wintertime NAO index and EOF2 time series (original figure 10d), and the similar spatial correlation patterns in the EOF2 vs. IB (original figure 10b) and wintertime NAO vs. IceBridge (Figure 1a below) correlation maps. A plot of the spatial correlation between IceBridge and annually averaged GBI (Figure 2 below; not in manuscript) strongly resembles the inverse of the spatial correlation between Ice-Bridge and annually averaged NAO (Figure 1b below), which is to be expected given the strong negative correlation between the annual NAO and GBI time series (Hanna et al., 2016). In the summertime, when the NAO and GBI show their largest differences (Hanna et al., 2016), we find a weak positive correlation between IceBridge accumulation and summertime GBI (Figure 1c below) – i.e. slightly higher accumulation during summers without overall enhanced blocking. While this may seem counter-intuitive, this relationship is driven by enhanced meridional flow and moisture advection into Greenland under the weak zonal flow associated with GBI positive (NAO negative) conditions (Hanna et al., 2016). Hanna et al. (2016) similarly find enhanced precipitation in central-northern Greenland associated with positive GBI summers (their Figure 6g). They also show negative precipitation anomalies in southeast Greenland during positive GBI summers (their Figure 6g), but our IceBridge data has poor coverage in this region. Note, also, that whereas Hanna et al. (2016) compare summer precipitation to summer GBI, we are only able to correlate annual precipitation to summer GBI.

Comment: p.11, l.25 -> "used to validate THE study".

Response: The suggested changes have been made.

Comment: p.11, l.30 "we hypthesize that rising accumulation over most of the GIS interior since 1976 is related to an increasing AMO index" – rising accum. could equally well reflect changes in atmospheric circulation, e.g. a more meridional airflow on average - with more moisture laden south-westerly winds, affecting Greenland.

Response: We no longer discuss this recent accumulation rise because it is not statistically significant. The relevant figure (original Figure 11) has also been removed.

Comment: p.12, l.6: The Hanna et al. (2013) reference cited here should be for the IJOC paper referenced above, not the Nature paper - please amend.

Response: The suggested changes have been made.

Comment: p.13, l.6 : change "strongest" to "most strongly". References Box & Rinke 2003 paper has the authors' names repeated twice. Please add other author names (or et al.) of the Shepherd 2012 Science paper.

Response: The suggested changes have been made.

Comment: Table 1: add in the caption what the plus/minus figures represent.

Response: The suggested changes have been made.

Figure 1: Pearson correlation map between 1899-2014 IceBridge accumulation and epoch-averaged climate indices. Statistically significant (p < 0.05) correlations are shown as larger data points. Maps show correlation of IceBridge data with a) Wintertime Jones (1997) NAO. b) Annual Jones (1997) NAO. c) Summer GBI. d) Annual AMO.

Figure 2: Pearson correlation map between 1899-2014 IceBridge accumulation and epoch-averaged annual GBI. Statistically significant (p < 0.05) correlations are shown as larger data points.
* * *
[Figure]

**Fig. 1.**

[Figure]

**Fig. 2.**